# Relative demographic susceptibility does not explain the extinction chronology of Sahul's megafauna

Corey JA Bradshaw[1,2]*, Christopher N Johnson[2,3], John Llewelyn[1,2], Vera Weisbecker[2,4], Giovanni Strona[5], Frédérik Saltré[1,2]

[1]Global Ecology *Partuyarta Ngadluku Wardli Kuu*, College of Science and Engineering, Flinders University, Tarndanya (Adelaide), Australia; [2]ARC Centre of Excellence for Australian Biodiversity and Heritage, Wollongong, Australia; [3]Dynamics of Eco-Evolutionary Pattern, University of Tasmania, Hobart, Australia; [4]College of Science and Engineering, Flinders University, Adelaide, Australia; [5]Faculty of Biological and Environmental Sciences, University of Helsinki, Helsinki, Finland

**Abstract** The causes of Sahul's megafauna extinctions remain uncertain, although several interacting factors were likely responsible. To examine the relative support for hypotheses regarding plausible ecological mechanisms underlying these extinctions, we constructed the first stochastic, age-structured models for 13 extinct megafauna species from five functional/taxonomic groups, as well as 8 extant species within these groups for comparison. Perturbing specific demographic rates individually, we tested which species were more demographically susceptible to extinction, and then compared these relative sensitivities to the fossil-derived extinction chronology. Our models show that the macropodiformes were the least demographically susceptible to extinction, followed by carnivores, monotremes, vombatiform herbivores, and large birds. Five of the eight extant species were as or more susceptible than the extinct species. There was no clear relationship between extinction susceptibility and the extinction chronology for any perturbation scenario, while body mass and generation length explained much of the variation in relative risk. Our results reveal that the actual mechanisms leading to the observed extinction chronology were unlikely related to variation in demographic susceptibility per se, but were possibly driven instead by finer-scale variation in climate change and/or human prey choice and relative hunting success.

*For correspondence:
corey.bradshaw@flinders.edu.au

**Competing interests:** The authors declare that no competing interests exist.

## Introduction

The myriad mechanisms driving species extinctions (*Sodhi et al., 2009*) are often synergistic (*Brook et al., 2008*), spatially variable (*Saltré et al., 2019*), phylogenetically clumped (*Fritz and Purvis, 2010*), correlated with population size (*O'Grady et al., 2004*), and dependent on biotic interactions (*Strona and Bradshaw, 2018*). This complexity means that even in contemporary settings involving closely monitored species, identifying the ecological mechanisms underlying the causes of a particular extinction can be difficult (*Caughley and Gunn, 1996*; *Fagan and Holmes, 2006*). This challenge is considerably greater for palaeo-extinctions because of the restricted ecological knowledge of extinct species. In the case of prehistoric extinctions, we can only infer the environmental conditions likely operating at the estimated time of the events from rare and sparsely distributed proxies (*Johnson et al., 2016*).

The rapid and widespread disappearance of megafauna in the late Quaternary on most continents is one of the best-studied extinction events of the past. Many plausible causes of megafaunal extinction have been proposed (*Koch and Barnosky, 2006*). The main drivers of these extinctions appear to differ depending on taxon, region, and time period (*Lorenzen et al., 2011*; *Metcalf et al., 2016*; *Wan and Zhang, 2017*), but there is growing consensus that multiple drivers were involved, including the interactions between climatic shifts and novel human pressure as dominant mechanisms (*Bartlett et al., 2016*; *Johnson et al., 2016*; *Saltré et al., 2019*; *Sandom et al., 2014*; *Villavicencio et al., 2016*). This consensus mostly relies on approaches examining extinction chronologies relative to indices of temporal and spatial environmental variation. While such correlative approaches can suggest potential causes of extinction, they cannot by themselves provide strong inference on the plausible ecological processes involved. Instead, approaches that construct mechanistic models of environmental and other processes that drive extinctions could reveal the relative susceptibility of species over the course of a large extinction event (*Alroy, 2001*; *Timmermann, 2020*). Revealing such mechanisms not only provides evidence-based explanations of how and why past extinction events unfolded, it also assists contemporary and future ecological analyses by describing extinction processes over longer timeframes than historical records permit. This can further contextualise baselines for conservation and management targets, and assist in predicting the magnitude and sequences of future extinctions (*Pardi and Smith, 2012*; *Willis and Birks, 2006*; *Wingard et al., 2017*) as the current crisis unfolds (*Bradshaw et al., 2021*).

Existing mechanistic models applied to megafauna systems differ in their complexity, ranging from predator-prey models (*Frank et al., 2015*; *Nogués-Bravo et al., 2008*), to fully age-structured stochastic models (*Prowse et al., 2013*), or stochastic predator-prey-competition functions (*Prowse et al., 2014*). If sufficiently comprehensive, such models can be useful tools to compare the likelihood of the processes driving extinctions, even in the deeper past. Although measuring the demographic rates of long-extinct species is impossible, robust estimates can be approximated from modern analogues and allometric relationships derived from extant species (*Prowse et al., 2014*; *Prowse et al., 2013*). This makes it possible to construct stochastic demographic models of both extinct and related extant species, and compare their relative susceptibility to perturbations by mimicking particular environmental processes *in silico*.

Despite these methodological advances, unravelling the causes of the disappearance of megafauna from Sahul (the combined landmass of Australia and New Guinea joined during periods of low sea level) is still a major challenge. This is because of the event's antiquity (*Johnson et al., 2016*) and the sparse palaeo-ecological information on megafauna extinctions compared to other parts of the world (*Johnson et al., 2016*; *Saltré et al., 2019*). However, based on the expectation that if high demographic susceptibility is an important feature of a species' actual extinction dynamics, the most susceptible species should have gone extinct before more resilient species did.

Stochastic demographic models can quantify the relative support for the hypothesized mechanisms potentially involved in the megafauna disappearances in Sahul (summarized in *Figure 1*), which could assist in explaining the observed extinction chronology. These mechanisms include: (*i*) There is a life history pattern in which the slowest-reproducing species succumbed first to changing conditions (e.g., novel and efficient human hunting and/or climate change) (*Cardillo et al., 2005*; *Purvis et al., 2000*; *Sodhi et al., 2009*). This mechanism assumes that human hunting or climate change, even if non-selective, would differentially remove species that were more demographically sensitive to increased mortality (*González-Suárez et al., 2013*; *Johnson, 2002*). (*ii*) The most susceptible species were those whose life histories conferred the highest sensitivity to human hunting, such as species most sensitive to the loss of juveniles (*Brook and Johnson, 2006*). (*iii*) Bottom-up processes drove the extinctions, as supported by differences in the extinction timing between carnivores and their herbivore prey. Under this mechanism, prey-specialist carnivores should be more susceptible than their prey (i.e. because they depend on declining prey populations), whereas more generalist carnivores that could switch food sources would be less susceptible than their main prey (*Chamberlain et al., 2005*; *Ripple and Van Valkenburgh, 2010*). (*iv*) Species susceptible to temporal variation in climate succumbed before those most able to adapt to changing conditions. Under this mechanism, we expect the largest species — that is, those possessing traits associated with diet/habitat generalism (*Monaco et al., 2020*), physiological resilience to fluctuating food availability (*Herfindal et al., 2006*; *Morris and Doak, 2004*), high endurance, and rapid, efficient dispersal away from stressful conditions (*Johnson, 2006*) — would persist the longest in the face of

| mechanism | schematic | description | scenario | hypothesis | reference |
|---|---|---|---|---|---|
| **allometry**<br>- *life-history speed* |  | - species with slowest life histories succumb first | LH | slower-generation species have earlier extinction dates | *Cardillo et al., 2005; Johnson 2006; Purvis et al., 2000; Selwood et al., 2015; Sodhi et al., 2009;* |
| - *fecundity* |  | - lowest-fecundity species succumb first | $\downarrow F$ | lowest-fecundity species succumb first to reductions in fertility | |
| **exposure to hunting** |  | - species more sensitive to reduction in juvenile survival succumb first | $\downarrow S_j$ | species with highest susceptibility to hunting juveniles succumb first | *Brook and Johnson, 2006* |
| **degree of predator dependency on prey** |  | - generalist predators succumb only after large, abundant herbivore prey disappears (prey switching) | $\downarrow S$, $\downarrow$ind | higher offtake rates/reductions in all-ages survival affect herbivores before predators | *Ripple and Van Valkenburgh, 2010* |
| |  | - specialist predators succumb before main herbivore prey disappears | $\downarrow S$, $\downarrow$ind | higher offtake rates/reductions in all-ages survival affect predators before herbivores | *Purvis et al., 2000* |
| **climate-change susceptibility**<br>- *physiological susceptibility* |  | - physiologically resilient and/or high-dispersal species endure longest<br>- high temperatures increase frequency of catastrophes | $\uparrow$cat$_f$, $\uparrow$cat$_M$ | increasing susceptibility to increasing frequency & magnitude of catastrophes drives climate-sensitive species extinct first | *Herfindal et al. 2006; Johnson, 2006; Morris and Doak, 2004; Selwood et al., 2015* |
| - *population size* |  | - smaller populations (larger species) most susceptible | $\uparrow$cat$_f$ | | *Cardillo et al. 2005* |
| - *juvenile survival* |  | - heat stress in parents reduces neonatal survival; heat stress in young animals | $\uparrow$cat$_f$, $\uparrow$cat$_M$ | species with most susceptible juveniles to heat stress ($\downarrow$survival) extinct first | *Selwood et al., 2015* |
| **differential selection/ease of access by humans** |  | - species best able to avoid human predation endure longest | NA | alternative hypothesis (not explicitly tested) | *Broughton et al., 2011; Janis et al., 2020* |

**Figure 1.** Description of five dominant mechanisms by which megafauna could have been driven to extinction and the associated seven perturbation scenarios examined.

catastrophic environmental change, independent of the intensity of human predation or changing climates. However, there are many other ways that climate change can alter demography (reviewed in *Selwood et al., 2015*). For example, heat stress can disadvantage juveniles by reducing parent capacity to raise neonates, or by affecting relatively heat-intolerant juveniles directly; species with more-sensitive juveniles are therefore expected to go extinct before more heat-tolerant species (*Selwood et al., 2015*). An increasing frequency of climate-induced catastrophes can also drive relatively smaller populations toward extinction faster, meaning large-bodied species with smaller populations are potentially more susceptible (*Cardillo et al., 2005*; *Liow et al., 2008*; *Tomiya, 2013*). (*v*) If none of the aforementioned mechanisms explains the extinction event's chronology, non-demographic mechanisms such as differential selection of or ease of access by human hunters could have played more important roles.

To examine the relative support of these hypotheses, we developed the first stochastic, age-structured demographic models ever constructed for 13 extinct megafauna species in Sahul broadly categorized into five functional/taxonomic groups: (*i*) 4 vombatiform herbivores, (*ii*) 5 macropodiform herbivores, (*iii*) 1 large, flightless bird, (*iv*) 2 marsupial carnivores, and (*v*) 1 monotreme invertivore. We also built demographic models for 8 of some of the largest, local, extant species, including representatives from each of the functional/taxonomic groups described above for comparison. Our null hypothesis is that these extant species should demonstrate higher resilience to perturbations than the extinct species, given that they persisted through the main extinction event to the present. Subjecting each species' model stochastically to different types of demographic perturbations, we tested seven scenarios (described in more details in Materials and methods) regarding the processes that could lead to extinction (see also *Figure 1*): an allometric relationship between the time of extinction and species' body mass and/or generation length (Scenario 'LH'), reducing juvenile survival (Scenario '$\downarrow S_j$') (*Brook and Johnson, 2006*; *Munn and Dawson, 2001*), reducing fertility (Scenario '$\downarrow F$') (*Gittleman and Thompson, 1988*; *Miller et al., 2016*; *Oftedal, 1985*), reducing survival across all ages (Scenario '$\downarrow S$'), individual offtake from the population via hunting (Scenario '$\downarrow$ind'), increasing environmental variability generating extreme climate events causing catastrophic mortality (Scenario '$\uparrow cat_f$'), and increasing the magnitude of environment-driven, catastrophic mortality events (Scenario '$\uparrow cat_M$') (*Reed et al., 2003*).

We hypothesize that one, or several, of these types of perturbations would provide a better match than the others between relative demographic susceptibility and the continental-scale chronology of extinctions as inferred from the fossil record. Identifying which, if any, of the scenarios best matches the chronology would therefore indicate higher relative support for those mechanisms being the most likely involved in driving the observed extinctions. We first compared the expectation of larger (and therefore, slower life-history; Scenario LH, *Figure 1*) species more likely to go extinct than smaller species when faced with novel mortality sources (*Brook and Bowman, 2005*), followed by the outcomes of all other scenarios (*Figure 1*) to test if sensitivity to specific demographic changes supported other mechanisms.

## Results

There was no indication that relatively heavier (*Figure 2a*) or slower life history (longer-generation; *Figure 2b*) species went extinct before lighter, faster life-history species (Scenario LH), even considering that the two mid- and small-sized carnivores *Thylacinus* and *Sarcophilus* went extinct on the mainland late in the Holocene at approximately the same time (~3200 years before present; *Figure 2*; *White et al., 2018*).

To distil overall extinction risk per species in each scenario, we progressively increased the relevant perturbation and calculated the proportion of 10,000 stochastic model runs where the final population size fell below a quasi-extinction threshold ($E_q$) of 50 females (*Frankham et al., 2014*). The area under the resulting quasi-extinction-probability curves provides a scenario-specific representation of extinction risk across the entire range of the specific perturbation. The quasi-extinction curves for each species differed markedly in each perturbation scenario (*Appendix 6—figure 1*), although there were some similarities among scenarios. For example, in all scenarios except for fertility reduction (Scenario $\downarrow F$) and offtake (Scenario $\downarrow$ind), the smallest, extant carnivore *Dasyurus* was the least susceptible, whereas *Genyornis* was one of the most susceptible in four of the six scenarios (*Appendix 6—figure 1*).

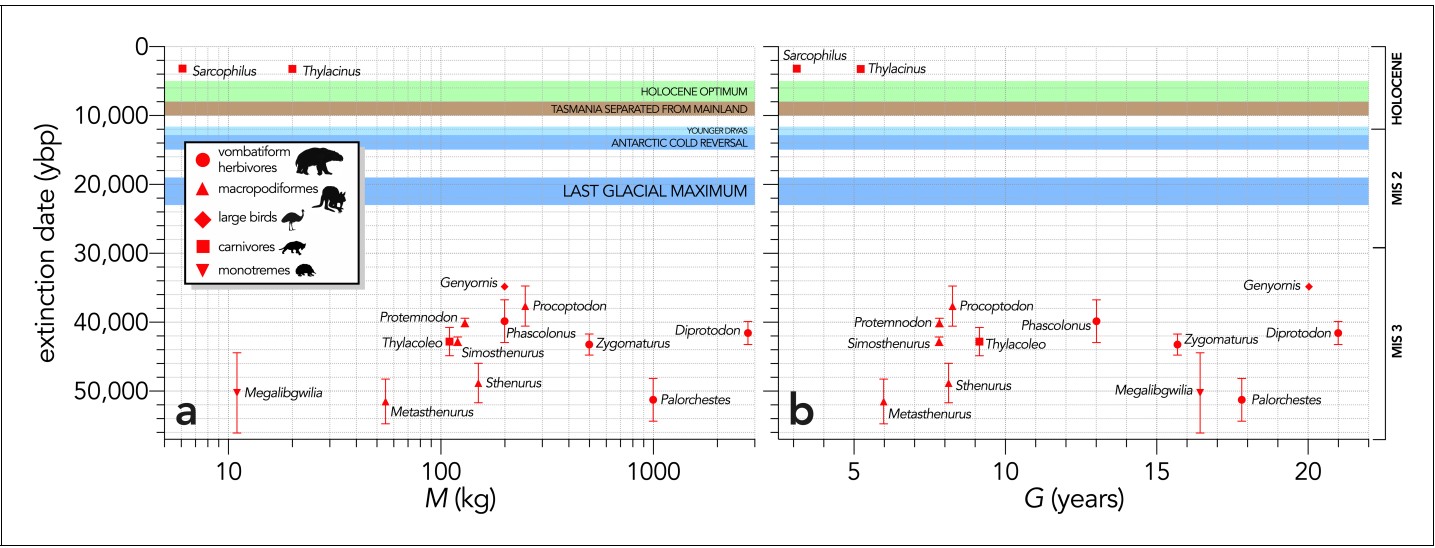

**Figure 2.** Relationship between estimated date of species extinction (across the entire continent) and (**a**) body mass (*M*, kg) or (**b**) generation length (*G*, years) (Scenario LH). Extinction-timing windows are estimated based on the agreement among six different models that correct for the Signor-Lipps effect (described in Materials and methods) in chronologies of quality-rated (*Rodríguez-Rey et al., 2015*) fossil dates for the studied taxa described in *Peters et al., 2019*. Here, we have depicted *Sarcophilus* as 'extant', even though it went extinct on the mainland >3000 years ago. Also shown are the approximate major climate periods and transitions: Marine Isotope Stage 3 (MIS 3), MIS 2 (including the Last Glacial Maximum, Antarctic Cold Reversal, and Younger Dryas), and the Holocene (including the period of sea level flooding when Tasmania separated from the mainland, and the relatively warm, wet, and climatically stable Holocene optimum).

Across all species, $\log_{10}$ body mass explained some of the variance in the total area under the quasi-extinction curve (*Figure 3*) for the individual-removal (↓ind: evidence ratio [ER]=41.22, $R^2 = 0.35$; *Figure 3d*) and catastrophe-magnitude (↑cat$_M$: ER = 19.92, $R^2 = 0.31$; *Figure 3f*) scenarios, but less variance for the scenarios with reductions in juvenile (↓$S_j$: ER = 2.04, $R^2 = 0.14$; *Figure 3a*) and all-ages survival (↓$S$: ER = 5.05, $R^2 = 0.21$; Scenario *iv*; *Figure 3c*). There was little to no evidence for a relationship in the fertility-reduction (↓$F$) and catastrophe-frequency (↑cat$_f$) scenarios (ER ≤ 1; *Figure 3b,e*). The relationships between area under the quasi-extinction curve and $\log_{10}$ generation length (*G*) were generally stronger (*Figure 4*). The strongest relationships here were for all-ages survival-reduction (↓$S$) and magnitude-of-catastrophe (↑cat$_M$) scenarios (ER >490, $R^2 ≥0.49$; *Figure 4c,f*), followed by weaker relationships (ER <11, $R^2 ≤0.26$) for Scenarios ↓$S_j$ (*Figure 4a*), ↓ind (*Figure 4d*), and ↑cat$_f$ (*Figure 4e*), and no evidence for a relationship in Scenario ↓$F$ (ER <1; *Figure 4b*).

Allometric scaling of extinction risk was also apparent within most taxonomic/functional groups. For example, *Diprotodon* had the highest extinction risk among the extinct vombatiform herbivores in every scenario except fertility reduction (Scenario ↓$F$; *Figures 3b* and *4b*). Most species were relatively immune even to large reductions in fertility, except for *Sarcophilus*, *Dasyurus*, *Vombatus*, *Alectura*, *Dromaius*, and *Genyornis* (*Figures 3b* and *4b*). In the offtake sub-scenario where we removed only juvenile individuals (↓ind$_j$), the results were qualitatively similar to Scenario ↓$S_j$ where we progressively decreased juvenile survival (*Appendix 7—figure 1*), although the relative susceptibility for most species decreased from Scenario ↓$S_j$ to ↓ind$_j$. However, susceptibility increased for the extinct carnivores *Thylacinus* and *Thylacoleo*, and remained approximately the same for *Dasyurus* and *Notamacropus* (*Appendix 7—figure 1*).

In the fertility-reduction sub-scenario (↓$F_e$) where we progressively increased the mean number of eggs removed per female per year in the bird species to emulate egg harvesting by humans, there was a progressively increasing susceptibility with body mass (*Figure 5*). *Genyornis* was clearly the most susceptible to extinction from this mechanism compared to the other two bird species (*Figure 4*).

When the extinction dates are viewed relative to the extinction risk (quasi-extinction integrals) calculated for each scenario, there is no indication that the most susceptible species went extinct earlier in any perturbation scenario (*Figure 6*). Taking the sum of the quasi-extinction integrals across

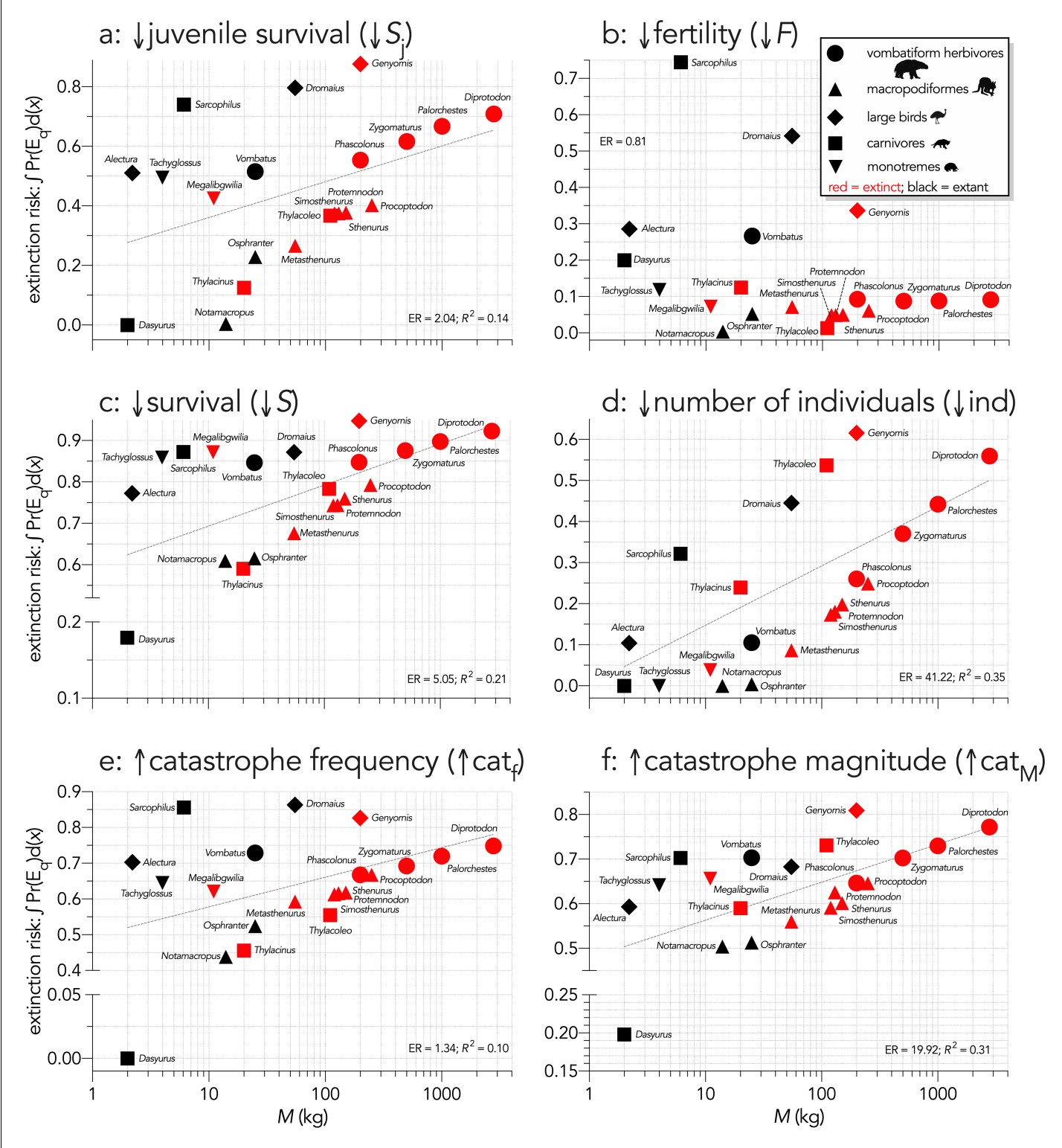

**Figure 3.** Area under the quasi-extinction curve (from *Appendix 6—figure 1*) — extinction risk: ∫Pr(E_q)d(*x*) — as a function of body mass (*M*, kg) for (**a**) (Scenario ↓*S*_j) decreasing juvenile survival, (**b**) (Scenario ↓*F*) decreasing fertility, (**c**) (Scenario ↓*S*) decreasing survival across all age classes, (**d**) (Scenario ↓ind) increasing number of individuals removed year$^{-1}$, (**e**) (Scenario ↑cat_f) increasing frequency of catastrophic die-offs per generation, and f: (Scenario ↑cat_M) increasing magnitude of catastrophic die-offs. Shown are the information-theoretic evidence ratios (ER) and variation explained (*R*$^2$) for the lines of best fit (grey dashed) in each scenario. Here, we have depicted *Sarcophilus* as 'extant', even though it went extinct on the mainland >3000 years ago.

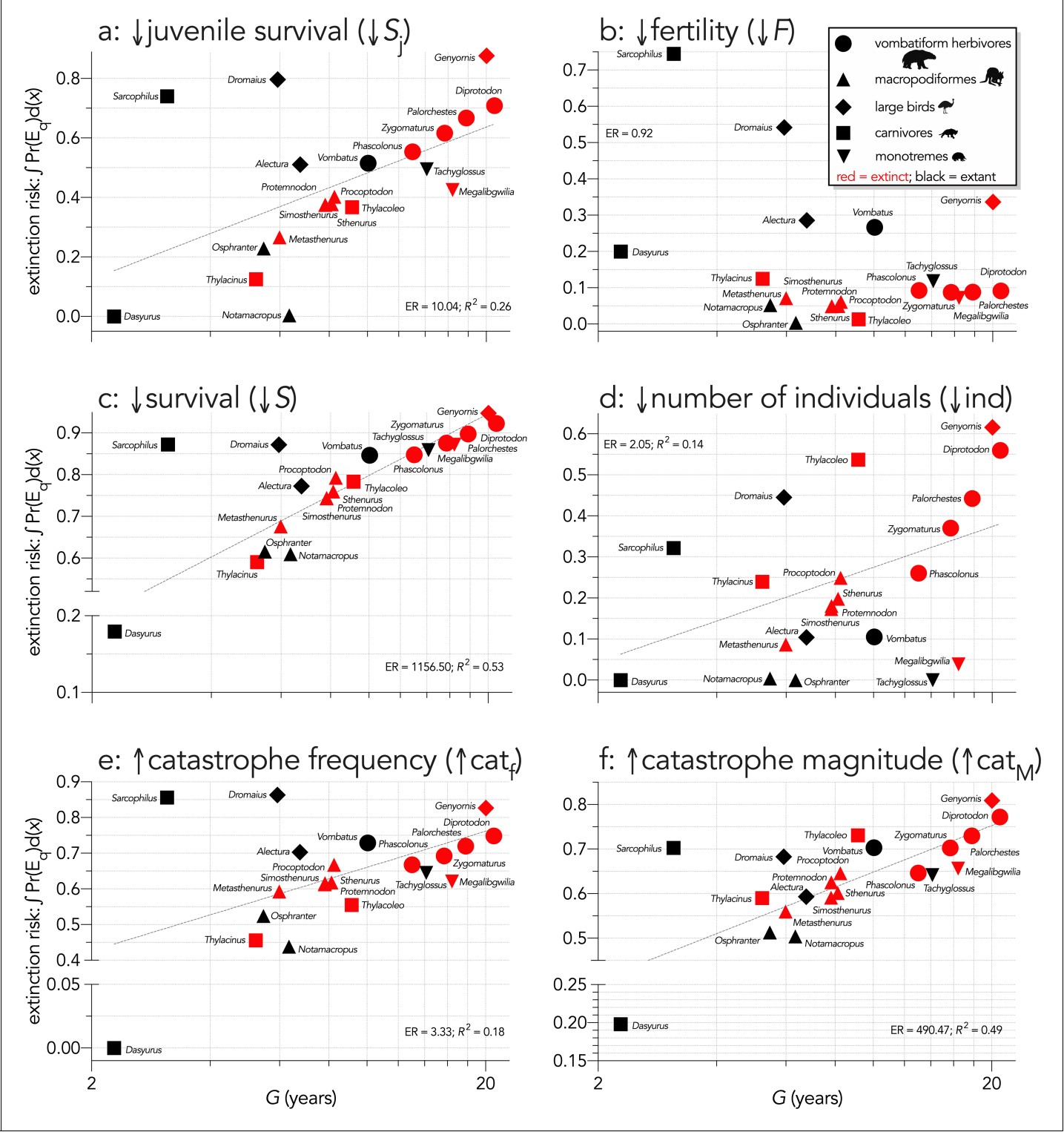

**Figure 4.** Area under the quasi-extinction curve (from *Appendix 6—figure 1*) — extinction risk: ∫Pr(E_q)d(x) — as a function of generation length (*G*, years) for (**a**) (Scenario ↓*S*_j) decreasing juvenile survival, (**b**) (Scenario ↓*F*) decreasing fertility, (**c**) (Scenario ↓*S*) decreasing survival across all age classes, (**d**) (Scenario ↓ind) increasing number of individuals removed year$^{-1}$, (**e**) (Scenario ↑cat_f) increasing frequency of catastrophic die-offs per generation, and f: (Scenario ↑cat_M) increasing magnitude of catastrophic die-offs. Shown are the information-theoretic evidence ratios (ER) and variation explained (*R*$^2$) for the lines of best fit (grey dashed) in each scenario. Here, we have depicted *Sarcophilus* as 'extant', even though it went extinct on the mainland >3000 years ago.

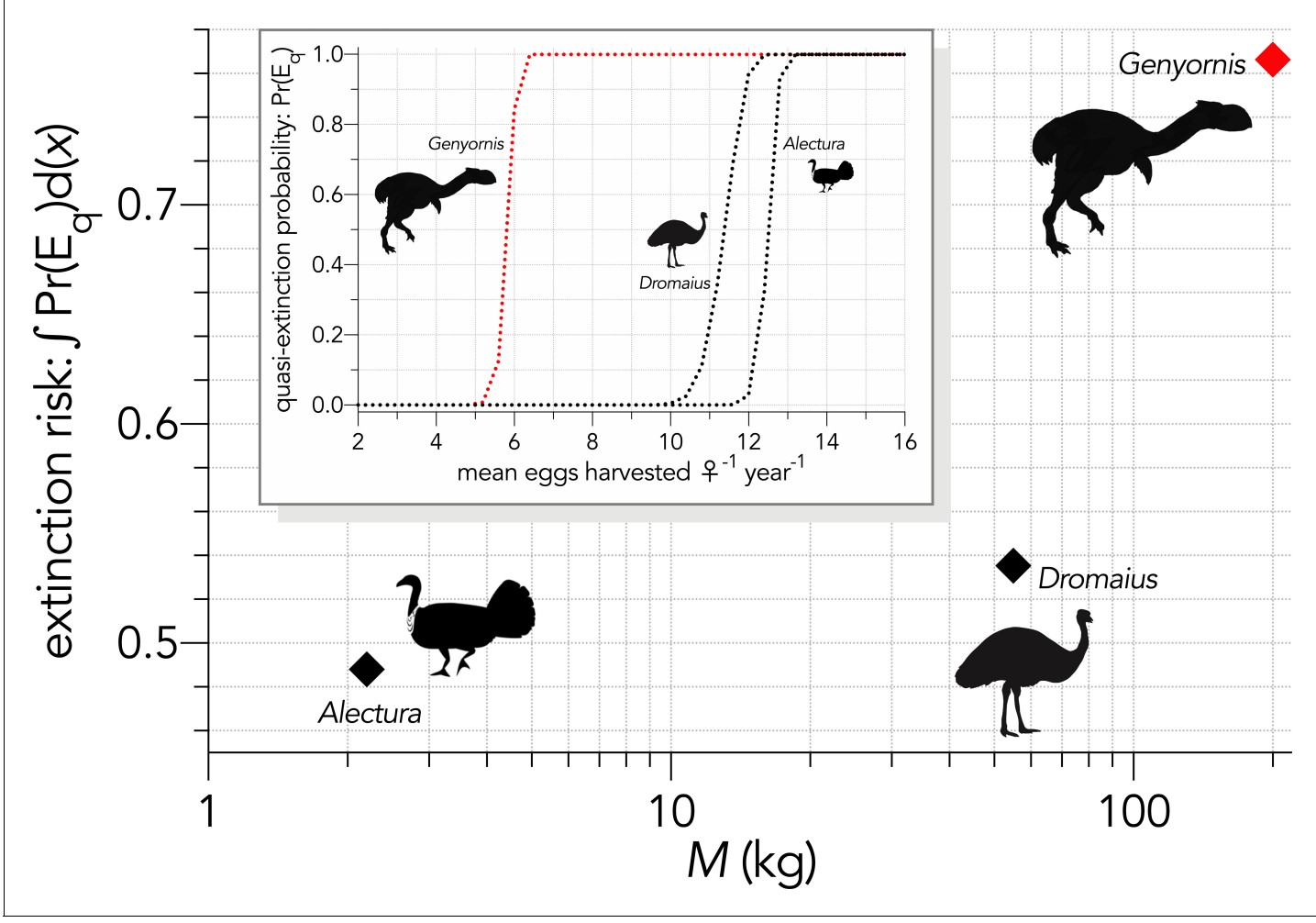

**Figure 5.** Inset: Increasing extinction risk for birds — quasi-extinction probability: Pr($E_q$) — as a function of increasing the mean number of eggs harvested per female per year (Scenario ↓$F_e$). The main graph shows the area under the quasi-extinction curve — extinction risk: $\int Pr(E_q)d(x)$ — as a function body mass (*M*, kg).

scenarios indicated that five of the eight extant species examined (*Sarcophilus* [extinct on mainland; extant in Tasmania], *Dromaius*, *Alectura*, *Vombatus*, *Tachyglossus*) had extinction risks that were equivalent or higher than most of the extinct species (*Figure 7a*). Taking the median rank of the quasi-extinction integral across scenarios generally indicated the lowest susceptibility in the macro-podiformes (although the small, extant carnivore *Dasyurus* was consistently the least susceptible for all scenarios except fertility reduction; *Figure 7a*), followed by the carnivores (except *Dasyurus*), monotremes, vombatiform herbivores, and finally, large birds (*Figure 7b*). The carnivores had sus-ceptibility ranks spread across most of the spectrum (*Figure 7b*).

Expressed as a function of $\log_{10}$ generation length, there was evidence for a moderate relation-ship across all species (ER = 12.07, $R^2$ = 0.27; *Figure 7c*), but removing the outlier species *Sarcophi-lus* and *Dromaius* resulted in a much stronger relationship (ER = 3.61 × $10^5$, $R^2$ = 0.76; *Figure 7c*). Susceptibility also tended to increase with body mass within a group, except for carnivores (*Sarcoph-ilus* being the anomaly) and monotremes (*Figure 7c*). There was no indication of a pattern when extinction date is plotted against median susceptibility rank (*Figure 7d*).

## Discussion

The megafauna species of Sahul demonstrate demographic susceptibility to extinction largely follow-ing expectations derived from threat risk in modern species — species with slower life histories have

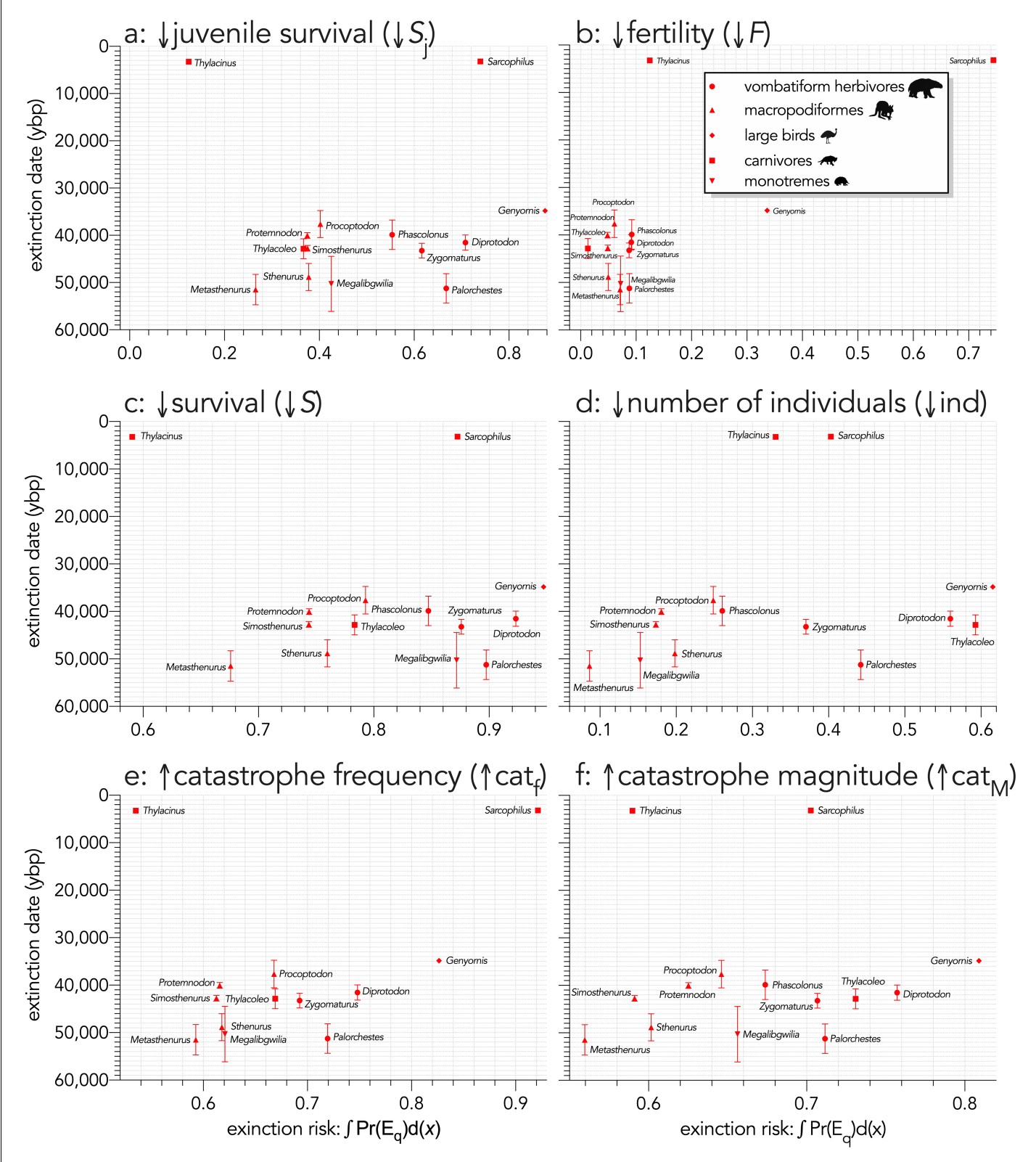

**Figure 6.** Relationship between estimated date of species extinction (across entire mainland) and area under the quasi-extinction curve (from *Appendix 7—figure 1*) — extinction risk: $\int Pr(E_q)d(x)$ — for (**a**) (Scenario $\downarrow S_j$) decreasing juvenile survival, (**b**) (Scenario $\downarrow F$) decreasing fertility, (**c**) (Scenario $\downarrow S$) decreasing survival across all age classes, (**d**) (Scenario $\downarrow$ind) increasing number of individuals removed year$^{-1}$, (**e**) (Scenario $\uparrow$cat$_f$) increasing frequency of catastrophic die-offs per generation, and f: (Scenario $\uparrow$**cat$_M$**) increasing magnitude of catastrophic die-offs. Extinction-timing

*Figure 6 continued on next page*

windows are estimated based on the agreement among six different models that correct for the Signor-Lipps effect (described in Materials and methods) in chronologies of quality-rated (*Rodríguez-Rey et al., 2015*) fossil dates for the studied taxa described in *Peters et al., 2019*.

higher demographic risk to extinction on average (*Cardillo et al., 2005*; *Liow et al., 2008*; *Purvis et al., 2000*; *Tomiya, 2013*). Indeed, our models show a convincing relationship between a taxon's overall relative demographic susceptibility to extinction and its generation length. While demography clearly must have played a role in the extinction of Sahul's megafauna given that mainly

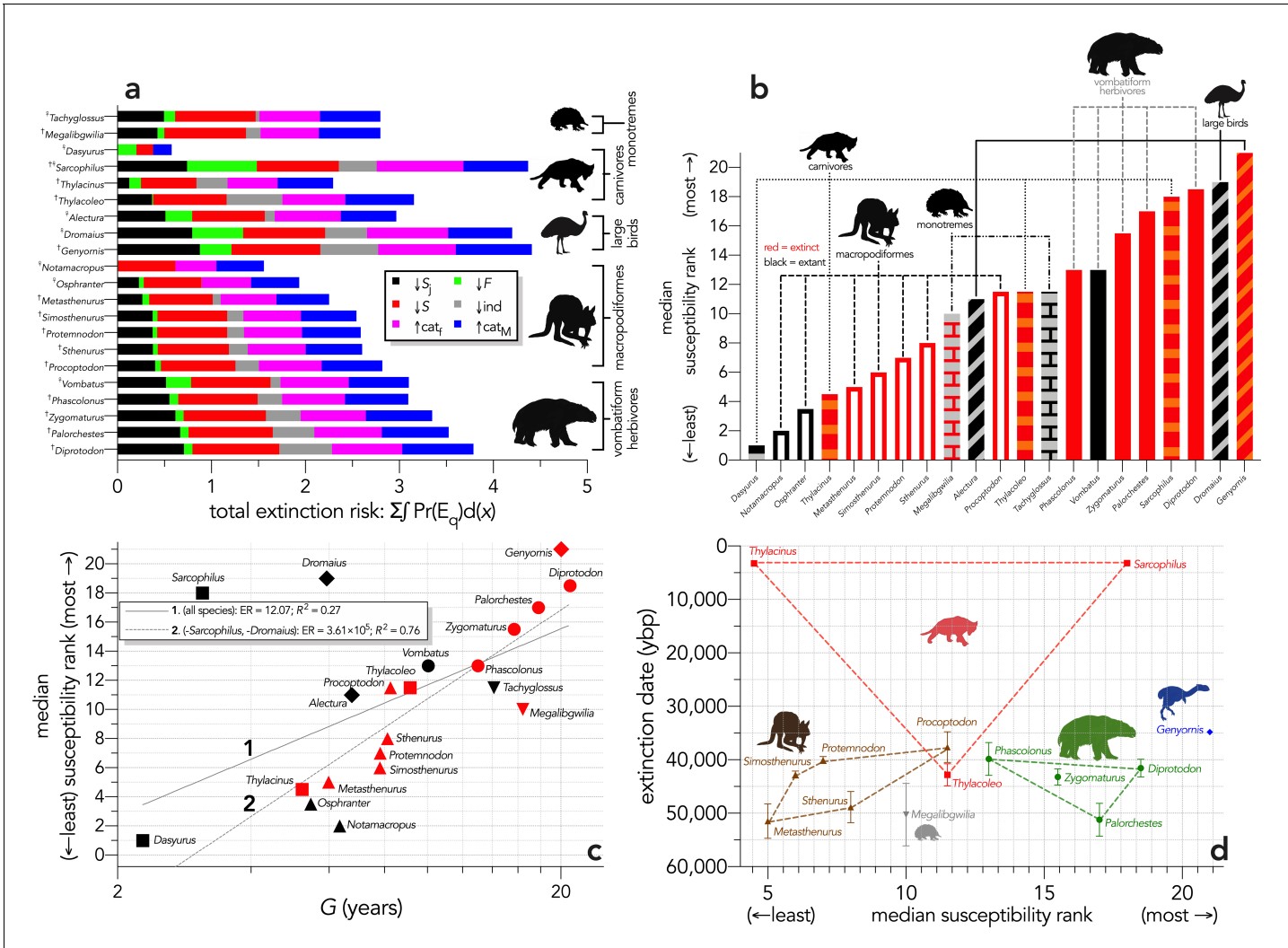

**Figure 7.** Extinction susceptibility of the 21 modelled species. (**a**) Sum of the areas under the quasi-extinction curve for each of the six scenarios considered — total extinction risk: $\Sigma \int \mathrm{Pr}(E_q)d(x)$ — for each of the 21 modelled species ([†]extinct; [♀]extant; scenario abbreviations: $\downarrow S_j$ = reducing juvenile survival; $\downarrow F$ = reducing fertility; $\downarrow S$ = reducing survival; ind = reducing number of individuals; $\uparrow cat_f$ = increasing frequency of catastrophe; $\uparrow cat_M$ = increasing magnitude of catastrophe); (**b**) median susceptibility rank across the six scenarios considered (where higher ranks = higher susceptibility to extinction) for each species (red = extinct; black = extant; outline-only bars = macropodiformes; solid bars = vombatiforms; angled crosshatching = birds; vertical crosshatching = carnivores; brick crosshatching = monotremes); (**c**) median susceptibility rank as a function of $\log_{10}$ generation length (*G*, kg) — there was a weak correlation including all species (solid grey line 1), but a strong relationship removing *Sarcophilus* and *Dromaius* (dashed gray line 2) (information-theoretic evidence ratio [ER] and variance explained [$R^2$] shown for each); (**d**) estimated date of species extinction (across entire continent) as a function of median susceptibility rank; taxonomic/functional groupings are indicated by coloured symbols and convex hulls (macropodids: brown; monotremes: grey; vombatiforms: green; birds: blue; carnivores: red). Extinction-timing windows are estimated based on the agreement among six different models that correct for the Signor-Lipps effect (described in Materials and methods) in chronologies of quality-rated (*Rodríguez-Rey et al., 2015*) fossil dates for the studied taxa described in *Peters et al., 2019*.

the largest species succumbed (*Lyons et al., 2016*), our stochastic models revealed no clear relationship between relative demographic susceptibility and the order in which the taxa we considered went extinct (even after considering alternative approaches to calculate the window of extinction — *Appendix 8—figure 1–3*). In particular, we did not find that the most demographically susceptible species went extinct before more resilient species. We can only conclude that the actual extinction chronology must have instead been an emergent property of many interacting demographic rates, temporal and spatial variation in population abundance, particular environmental contexts, community interactions, and likely other traits (*Sodhi et al., 2009*). As different perturbations compromise different aspects of a species' life history, its relative susceptibility to extinction compared to other species in its community varies in often unpredictable ways.

We can therefore reject the hypothesis that the continent-wide extinction chronology is explained by species' relative demographic susceptibility and non-selective hunting by humans. If demographic susceptibility coupled with non-selective hunting were the primary causes of these extinctions, we would expect a relationship between vulnerability and extinction date whereby the more vulnerable species went extinct earlier — but we found no such relationship (*Figure 7d*). However, the comparison of susceptibility under increasing intensities of egg harvesting revealed the highest demographic risk from this type of activity for the extinct *Genyornis* compared to the extant *Dromaius* and *Alectura* birds, supporting the notion that egg harvesting (and not hunting of adults) might have been at least partially responsible for the demise of *Genyornis* (*Miller et al., 2016*). We can also reject the hypothesis that the largest, and therefore the most physiologically buffered and mobile species, were the most resilient given the lack of relationship with the inferred extinction chronology (*Figures 2* and *7d*). Neither did the species with the highest sensitivities to reductions arising from the various perturbation scenarios succumb earlier (*Figures 5* and *6d*).

This lack of relationship to the chronology, combined with the result that many of the extant species had some of the highest extinction susceptibilities, suggest that no obvious demographic properties can explain the taxon-specific timing within the Sahul extinction event of the Late Pleistocene. This opens the possibility that the chronology instead reflects either a random set of circumstances — that species succumbed to circumstantial combinations of stressors depending on the local perturbations experienced by particular populations (*Saltré et al., 2019*) — or even that the chronology is still insufficiently resolved. However, this conclusion does not accord well with the notion that Late Pleistocene megafauna extinctions were non-random and occurred at a much higher pace than background extinction rates (*Johnson, 2006*; *Koch and Barnosky, 2006*).

Another possibility is that in the case of human hunting, preferences for selecting or avoiding particular species, such as targeting larger species for more efficient returns (*Broughton et al., 2011*), could have overridden or interacted with intrinsic demographic susceptibility. Indeed, *Lyons et al., 2016* concluded that either large-bodied mammals were selectively targeted by humans during the Late Quaternary, or that these species were relatively more vulnerable to human hunting than smaller-bodied species, or both. In addition, specific behavioural adaptations could potentially have rendered demographically high-risk species in fact *less* vulnerable to human hunting, such as the behavior of *Vombatus* to dig and defend burrows that were difficult to access by humans (*Garvey et al., 2016*) compared to larger burrowing or non-burrowing vombatiformes. In the case of the macropodiformes, interspecific variation in the type of locomotion — a trait not captured by our demographic models — could have contributed more to their relative susceptibility to human hunters. For example, the ability to hop at high velocities as in *Osphranter rufus* could have given it an escape advantage over the relatively slower sthenurine macropodiformes that likely employed more bipedal striding than hopping (*Janis et al., 2020*). Similar hypotheses regarding risk-persistence mismatches in multispecies simulations have been proposed for the Late Pleistocene megafauna extinction event in North America (*Alroy, 2001*).

Although marsupials are widely included in studies estimating the types of mammalian demographic relationships like those we used here (*de Magalhães et al., 2007*; *Healy et al., 2014*; *McCarthy et al., 2008*), more explicit consideration of their reproductive differences compared to placentals might further improve the resolution of future models. In particular, marsupials are born at the extreme altricial stage and complete most of their development *ex utero* through lactation (*Tyndale-Biscoe, 2005*). This might change the way the cost of reproduction is borne, because the unusually long period of marsupial lactation can reduce the cost of raising offspring per unit time (*Cork and Dove, 1989*; *Weisbecker and Goswami, 2010*). There is also an increasing number of

small marsupial and placental mammals known to have gone extinct during the Pleistocene in Sahul (e.g. *Cramb et al., 2009*; *Cramb et al., 2018*). The inclusion of these smaller species in future demographic susceptibility/chronology of extinction analyses will likely be insightful, but unfortunately estimates of extinction dates — which require at least 8–10 reliability dated specimens (*Bradshaw et al., 2012*; *Saltré et al., 2015*) — are not yet available for these animals.

Our models, although age-structured, stochastic, and incorporating compensatory density feedback, are still simplified expressions of a species' particular ecological and environmental contexts. As stated, our models are aspatial, yet we know that spatial processes are correlated with local extinctions across the landscape (*Saltré et al., 2019*). For example, large proboscideans like mammoths managed to persist well into the Holocene on island refugia despite having a high intrinsic extinction risk (*Nogués-Bravo et al., 2008*). It is therefore plausible that more localized measures of extinction risk, timing, and particular climate and habitat contexts (see Appendix 9 for an examination of demographic susceptibility relative to hindcasted climate trends) could reveal subtler demographic processes at work (*Chase et al., 2020*). However, Sahul's fossil record is still generally too sparse at a regional level to test this properly (*Peters et al., 2019*; *Rodríguez-Rey et al., 2016*), nor do we have data indicating how spatial variation might have altered local expressions of demographic rates in long-extinct species.

Our models also ignore biotic dependencies such as predator-prey, plant-herbivore, and competition relationships that could have modified relative susceptibility in different ways depending on the community in question (*Brook and Bowman, 2004*; *Choquenot and Bowman, 1998*). Trophic community networks constructed for south-eastern Sahul show that bottom-up processes most strongly affect lower trophic levels, with their influence diminishing at higher trophic levels, although extinct carnivores were more vulnerable to coextinction than extant carnivores (*Llewelyn et al., 2020*).

The particulars of the *Genyornis* extinction are also still debatable given the possibility that the egg shells used to date the species (*Miller et al., 2005*) are potentially confounded with an extinct *Progura* megapode (*Grellet-Tinner et al., 2016*; *Shute et al., 2017*). However, removing '*Genyornis*' from the extinction chronology makes no difference to our overall conclusions, but it is problematic for comparing relative extinction risk between the extinct *Genyornis* and the extant *Dromaius* and *Alectura*. In fact, by including an extant megapode (*Alectura*) in our model simulations, we determined that this much smaller (2.2 kg) and faster-reproducing species had a much lower extinction susceptibility than both *Genyornis* and *Dromaius*.

That we found no clear patterns among the extinct megafauna of Sahul to explain their relative extinction chronology supports the notion that, at least for mammals, risk can be high across all body masses depending on a species' particular ecology (*Davidson et al., 2009*), even if relative extinction risk appears to follow allometric expectations (*Brook and Bowman, 2005*; *Cardillo et al., 2005*; *Liow et al., 2008*; *Purvis et al., 2000*; *Tomiya, 2013*) as we demonstrated clearly here (*Figures 3*, *5* and *7*). By definition, the megafauna were generally large (>45 kg) species, yet neither their body mass or their correspondingly higher relative demographic susceptibility explains the extinction chronology in Sahul. Our approach also provides a template for assessing relative demographic susceptibility to extinction for other Sahul species that we did not consider here, and for those in other continents, that could reveal previously underappreciated dynamics and drivers. Nonetheless, more spatially and community-dependent models are still needed to provide a more complete picture of the dynamics of Late Pleistocene megafauna extinctions.

## Materials and methods

### Choice of species

Our first step was to choose enough extinct and extant species from the Sahul fossil record (*Peters et al., 2019*; *Rodríguez-Rey et al., 2016*) to represent a diversity of clades that were particularly affected during the main extinction event (estimated between 60 and 40 ka, where one ka = 1000 years ago) (*Saltré et al., 2016*). We also aimed to include at least one extant species within each functional/taxonomic group to compare extant with extinct species' susceptibility. We settled on a total of 21 species (13 extinct; 8 extant) from five different functional/taxonomic groups: (*i*) 5 vombatiform herbivores, (*ii*) 7 macropodiform herbivores, (*iii*) 3 large birds, (*iv*) 4 carnivores, and

(*v*) 2 monotreme invertivores. For a full list and justification of species chosen as well as the distribution of mean body masses, refer to Appendix 1.

## Estimating demographic rates

To build age-structured population models for extinct taxa, we relied on different allometric, phylogenetic, and measured relationships to predict the plausible range of component demographic rates. For most extant marsupials, we relied mainly on the marsupial life-history database published in *Fisher et al., 2001*, but updated some values for some species with more recent sources (see below). A detailed description of how we estimated the necessary demographic rates and other ecological data to build the stochastic models is provided in Appendix 2, and a full table of all demographic values is provided in *Appendix 2—table 1*. We also provide a correlation matrix among demographic values across species (*Appendix 2—table 2*).

## Age-structured (Leslie) population models

From the estimated demographic rates for each species, we constructed a pre-breeding, $\omega+1$ (*i*) $\times$ $\omega+1$ (*j*) element (representing ages from 0 to $\omega$ years old), Leslie transition matrix (**M**) for females only (males are demographically irrelevant assuming equal sex ratios). Fertilities ($m_x$) occupied the first row of the matrix, survival probabilities ($S_x$) occupied the sub-diagonal, and we set the final diagonal transition probability ($\mathbf{M}_{i,j}$) to $S_\omega$ for all species except *Vombatus*, *Thylacinus*, and *Sarcophilus* for which we instead set the value to 0 to limit unrealistically high proportions of old individuals in the population, and the evidence for catastrophic mortality at $\omega$ for the latter two species (dasyurids) (*Cockburn, 1997*; *Holz and Little, 1995*; *Oakwood et al., 2001*). Multiplying **M** by a population vector **n** estimates total population size at each forecasted time step (*Caswell, 2001*). Here, we used $\mathbf{n}_0 = AD\mathbf{M}\mathbf{w}$, where **w** = the right eigenvector of **M** (stable stage distribution), and *A* = the surface area of the study zone applied in the stochastic extinction scenarios — we arbitrarily chose $A = 250,000$ km$^2$ (500 km $\times$ 500 km; approximately 10% larger than the state of Victoria) so that the species with the lowest $\mathbf{n}_0$ would have a population of at least several thousand individuals at the start of the simulations (see Appendix 3). We also included a compensatory density-feedback function in all simulations to avoid exponentially increasing populations (see Appendix 4).

## Stochastic extinction scenarios

With the base **M** including density feedback tailored for each species, we perturbed various elements of their life histories to examine the relative support for hypotheses regarding plausible extinction drivers and pathways (see *Figure 1*). We first tested the relationship between extinction date and speed of life history as a baseline without any perturbation (Scenario LH), and then we generated six additional scenarios with perturbations. The second scenario ($\downarrow S_j$) decreased juvenile ($x = 0$ to $\alpha-1$) survival (plus a sub-scenario [$\downarrow$ind$_j$] where we progressively removed individual juveniles from the population as we did for all individuals in Scenario $\downarrow$ind — see below). This scenario aims to emulate either food shortages of sufficient magnitude to make growing juveniles with higher relative energy and water demand (*Munn and Dawson, 2001*) succumb to environmental change more than adults, or from targeted hunting of juveniles by humans (*Brook and Johnson, 2006*). The third scenario ($\downarrow F$) progressively reduces fertility to emulate food shortages lowering energetically demanding reproduction/lactation (*Gittleman and Thompson, 1988*; *Oftedal, 1985*). We also considered a sub-scenario (Scenario $\downarrow F_e$, see details below) for the category of large birds where we progressively increased the number of eggs harvested by humans (*Miller et al., 2016*). In Scenario $\downarrow S$, we progressively reduced survival across all age classes to examine the influence of an age-independent environmental stressor. Scenario $\downarrow$ind progressively removed individuals from the **n** population vector emulating offtake where animals are directly removed from the population to simulate human hunting (with age-relative offtake following the stable stage distribution of the target species). In Scenario $\uparrow$cat$_f$, we emulated how environmental variability would compromise populations via an increased relative (i.e. per generation) frequency of catastrophic die-offs by progressively increasing the number of catastrophic ~50% mortality events occurring per generation. Finally, Scenario $\uparrow$cat$_M$ progressively increased the magnitude of the catastrophic mortality events to examine species' responses to rising severity of catastrophes (*Reed et al., 2003*).

For Scenario $\downarrow F_e$, we estimated the egg-production component for *Genyornis* by calculating the proportion of total fecundity contributed by individual egg production in *Dromaius* (nest success of 0.406 × hatching probability of 0.419 = 0.17), and then multiplying this proportion by the total fertility estimated for *Genyornis* from *Equation 11* — this produced an estimated per-individual annual egg production of 7.74 eggs for *Genyornis* (or, 7.74/2 = 3.87 eggs resulting in daughters). For Scenario $\uparrow cat_M$, we randomly allocated a multiplier of the expected frequency per generation (uniform sampling) derived from the species-specific range of multipliers identified in Scenario $\uparrow cat_f$ (i.e. from one to the value where the species has an extinction probability = 1). In this way, we both standardized the relative risk among species and avoided cases where catastrophe frequency was insufficient to elicit any iterations without at least one extinction.

We ran 10,000 stochastic iterations of each model starting with allometrically predicted stable population size (see Appendix 3) divided into age classes according to the stable stage distribution. We projected all runs to $40\lfloor G \rfloor$ for each species (removing the first $\lceil G \rceil$ values as burn-in). In each scenario, we progressively increased the relevant perturbation and calculated the proportion of 10,000 stochastic model runs where the final population size fell below a quasi-extinction ($E_q$) of 50 female individuals (100 total individuals total assuming 1:1 sex ratios). This threshold is based on the updated minimum size below which a population cannot avoid inbreeding depression (*Frankham et al., 2014*). After calculating the per-increment probability of $E_q$ in each of the seven scenarios, we calculated the total area under the quasi-extinction curve (integral) for each species as a scenario-specific representation of extinction risk across the entire range of the specific perturbation — this provides a single, relative value per species for comparison. Finally, we ranked the integrals among species in each scenario (lower ranks = higher resilience), and took the median rank as an index of resilience to extinction incorporating all scenario sensitivities into one value for each species.

## Extinction dates

We compared the relative susceptibilities among all extinction scenarios, as well as the combined extinction-susceptibility rank of each species, to estimates of continental extinction times for the genera we examined. We took all estimates of continental extinction dates from the Signor-Lipps corrected values provided in *Saltré et al., 2016*; however, more recent continent-wide disappearance dates for *Thylacinus* and *Sarcophilus* are provided in *White et al., 2018*. The Signor-Lipps-correction of extinction timing is explained in detail in *Saltré et al., 2016*, but we briefly summarize it here. The mean date of extinction and its confidence intervals are derived from six frequentist methods to infer the timing of extinction — *Strauss and Sadler, 1989*, *McCarthy, 1998*, *Marshall, 1997*, *Solow et al., 2006*, *McInerny et al., 2006*, and the Gaussian-Resampled, Inverse-Weighted McInerny (GRIWM) model (*Bradshaw et al., 2012*) — from chronologies of dated fossils described in the *FosSahul* 2.0 database (*Peters et al., 2019*). All dates in a taxon's chronology are first assessed for reliability, with dates of less than 'A' rejected for the calculation of the extinction window (*Rodríguez-Rey et al., 2015*). Each method calculates an extinction window for each species (taxon), and from these a window of cross-model agreement is calculated for every year (assuming that higher cross-model agreement indicates a greater likelihood of the true extinction date occurring during those times). We also examined the sensitivity of our overall results to uncertainty in extinction estimates by deriving a jack-knifed version of the GRIWM model (*Bradshaw et al., 2012*; *Saltré et al., 2015*) (Appendix 8).

We hypothesize that one, or several, of these types of perturbation scenarios would lead to a better match between the continental-scale chronology of extinctions as inferred from the fossil record compared to the simpler expectation of larger species with slower life-histories being more likely to go extinct than smaller species with faster life histories when faced with novel mortality sources (Scenario LH) (*Brook and Bowman, 2005*).

## Data and code availability

All data and are R code needed to reproduce the analyses are available for download at github. com/cjabradshaw/MegafaunaSusceptibility.

## Acknowledgements

This study was supported by the Australian Research Council through a Centre of Excellence grant (CE170100015) to CJAB, CNJ, and VW, and a Discovery Project grant (DP170103227) to VW. We acknowledge the Indigenous Traditional Owners of the land on which Flinders University is built — the Kaurna people of the Adelaide Plains.

## Additional information

### Funding

| Funder | Grant reference number | Author |
| --- | --- | --- |
| Australian Research Council Centre of Excellence for Australian Biodiversity and Heritage | CE170100015 | Corey JA Bradshaw<br>Christopher N Johnson<br>Vera Weisbecker |
| Australian Research Council | DP170103227 | Vera Weisbecker |

The funders had no role in study design, data collection and interpretation, or the decision to submit the work for publication.

### Author contributions

Corey JA Bradshaw, Conceptualization, Resources, Data curation, Software, Formal analysis, Funding acquisition, Investigation, Visualization, Methodology, Writing - original draft, Project administration, Writing - review and editing, C.J.A.B conceptualized and wrote the paper, designed the research, wrote the code, sourced the data, and did the analysis; Christopher N Johnson, Conceptualization, Resources, Methodology, Writing - review and editing, C.J. provided additional analytical approaches and data, and contributed to writing the manuscript; John Llewelyn, Conceptualization, Resources, Writing - review and editing; Vera Weisbecker, Giovanni Strona, Conceptualization, Writing - review and editing; Frédérik Saltré, Conceptualization, Resources, Software, Formal analysis, Methodology, Writing - review and editing, Conceptualised and assisted writing the paper, and provided climate data

### Author ORCIDs

Corey JA Bradshaw (iD) https://orcid.org/0000-0002-5328-7741
Christopher N Johnson (iD) https://orcid.org/0000-0002-9719-3771
John Llewelyn (iD) https://orcid.org/0000-0002-5379-5631
Vera Weisbecker (iD) https://orcid.org/0000-0003-2370-4046
Giovanni Strona (iD) https://orcid.org/0000-0003-2294-4013
Frédérik Saltré (iD) https://orcid.org/0000-0002-5040-3911

### Decision letter and Author response

Decision letter https://doi.org/10.7554/eLife.63870.sa1
Author response https://doi.org/10.7554/eLife.63870.sa2

## Additional files

### Supplementary files

• Transparent reporting form

### Data availability

All data and are R code needed to reproduce the analyses are available for download at http://github.com/cjabradshaw/MegafaunaSusceptibility.

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

# Appendix 1

## Choice of species and body mass distribution

Given data availability, we settled on a total of 21 species (13 extinct; 8 extant) from five different functional/taxonomic groups: (*i*) 5 vombatiform herbivores: *Diprotodon optatum* (2786 kg; extinct) (*Wroe et al., 2004*), *Palorchestes azael* (1000 kg; extinct) (*Richards et al., 2019*), *Zygomaturus trilobus* (500 kg; extinct) (*Johnson, 2006*), *Phascolonus gigas* (200 kg; extinct) (*Johnson, 2006*), and *Vombatus ursinus* (common wombat; 25 kg; extant) (*McIlroy, 1996*; *Saran et al., 2011*); (*ii*) 7 macropodiform herbivores: *Procoptodon goliah* (250 kg; extinct) (*Johnson and Prideaux, 2004*), *Sthenurus stirlingi* (150 kg; extinct) (*Johnson and Prideaux, 2004*), *Protemnodon anak* (130 kg; extinct) (*Johnson, 2006*), *Simosthenurus occidentalis* (120 kg; extinct) (*Johnson, 2006*), *Metasthenurus newtonae* (55 kg; extinct) (*Johnson, 2006*), *Osphranter rufus* (red kangaroo; 25 kg; extant) (*McIlroy, 2008*), and *Notamacropus rufogriseus* (red-necked wallaby; 14 kg; extant) (*Strahan, 1991*); (*iii*) 3 omnivorous (but primarily plant-eating) large birds: *Genyornis newtoni* (200 kg; extinct) (*Johnson, 2006*), *Dromaius novaehollandiae* (emu; 55 kg; extant) (*Sales, 2007*), and *Alectura lathami* (brush turkey; 2.2 kg; extant) (*Jones et al., 1995*); (*iv*) 4 carnivores: *Thylacoleo carnifex* (marsupial 'lion'; 110 kg; extinct) (*Johnson, 2006*), *Thylacinus cynocephalus* (marsupial 'tiger'; 20 kg; extinct) (*Jones and Stoddart, 1998*), *Sarcophilus harrisii* (devil; 6.1 kg; extinct in mainland Australia, but extant in Tasmania — see below) (*Guiler, 1978*), and *Dasyurus maculatus* (spotted-tail quoll; 2.0 kg; extant) (*Belcher et al., 2008*); and (*v*) 2 monotreme invertivores: *Megalibgwilia ramsayi* (11 kg; extinct) (*Johnson, 2006*), and *Tachyglossus aculeatus* (short-beaked echidna; 4.0 kg; extant) (*Nicol and Andersen, 2007*).

For each species, we identified the body mass of mature females. However, the sex of an extinct individual from its fossilized remains in many species is difficult to determine, especially when sample sizes are small (*Alonso-Llamazares and Pablos, 2019*). As such, we might have inadvertently assigned a female mass based on an estimated male mass, given evidence that there is a male bias in many fossil collections (*Allentoft et al., 2010*; *Gower et al., 2019*). For this reason, we attempted to cover a broad range of body masses among species to maximize the *relative* difference between them for comparison.

The two genera *Thylacinus* and *Sarcophilus* require special consideration in both the design of the analysis and the interpretation of the results. While *Sarcophilus* is extant, it is restricted to the island state of Tasmania that has been separated from mainland Australia since approximately 8–10 ka. However, the species went extinct on the mainland 3179 (±24) years ago, whether considering the species complex *Sarcophilus harrisii* and *S. laniarius* together or separately (*White et al., 2018*). Like *Sarcophilus*, *Thylacinus* went extinct on the mainland just over 3000 years ago (*White et al., 2018*) and persisted in Tasmania. However, unlike *Sarcophilus*, *Thylacinus* also went extinct in Tasmania in the 1930s. In our analyses we treat *Sarcophilus* as 'extant', with the proviso that it should be considered extinct on the mainland. Although we could also have treated *Thylacinus* as 'extant' in the sense that it persisted into historical times in Tasmania, we treat this species as extinct in our analyses.

The distribution of the body masses (*M*) across the 21 species (range: 1.68–2786 kg) was approximately log-Normal (Shapiro-Wilk Normality test on $\log_{10}M$: $W = 0.9804$; p=0.9305; *Appendix 1—figure 1*). Median dates of continental (i.e., total species) extinction ranged from 51,470 (±3167 standard deviation) for *Metasthenurus*, to 3179 (±24) for *Sarcophilus* (mainland only; currently extant in Tasmania) (*Saltré et al., 2016*; *White et al., 2018*).

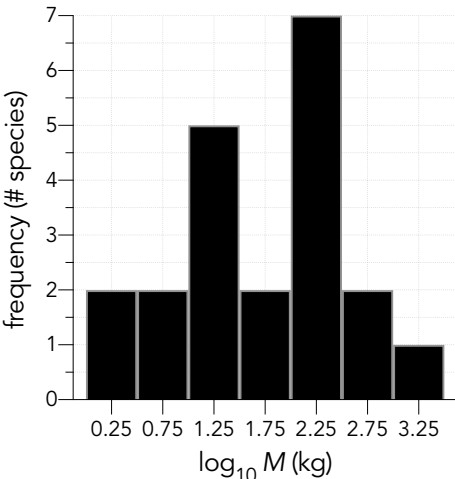

**Appendix 1—figure 1.** Histogram of $\log_{10}$ adult body masses (in kg) for the 21 species examined. The distribution is approximately log-Normal (Shapiro-Wilk Normality test on $\log_{10}M$: $W = 0.9804$; $p=0.9305$).

## Appendix 2

### Estimating demographic rates and population data as input parameters for the stochastic models

For each species, we first calculated the maximum rate of instantaneous population growth ($r_\mathrm{m}$) using the following equation:

$$r_\mathrm{m} = 10^{0.6914 - 0.2622 \log_{10} M} \tag{1}$$

for mammals, where $M$ = mass (g) (**Hennemann, 1983**). For birds, we used an optimization of the objective function based on age at first breeding ($\alpha$, estimated as shown below), adult survival ($s_\mathrm{ad}$, estimated as shown below), and the allometric constant for birds (**Dillingham et al., 2016**) $a_{rT}$ = 1.107:

$$r_\mathrm{m} = \log_e \lambda_1 = \log_e \left( \min \left| \lambda_1 - e^{\frac{a_{rT}}{\alpha + \frac{s_\mathrm{ad}}{\lambda_1 - s_\mathrm{ad}}}} \right| \right) \tag{2}$$

We then calculated theoretical equilibrium population densities ($D$, km$^{-2}$) based on the following:

$$D = 10^{4.196 - 0.740 \log_{10} M} / 2 \tag{3}$$

for mammalian herbivores ($M$ = body mass in g), where dividing by two predicts for females only (i. e., assumed 1:1 sex ratio) (**Damuth, 1981**). For large, flightless birds (i.e., *Genyornis* and *Dromaius*) (**Latham et al., 2020**), we applied:

$$D = 10^{3.65 - 0.82 \log_{10} M} / 2 \tag{4}$$

and:

$$D = 10^{1.63 - 0.23 \log_{10} M} / 2 \tag{5}$$

for omnivorous birds (i.e., *Alectura*) (**Juanes, 1986**) where $M$ is in g. For mammalian carnivores, we applied:

$$D = e^{1.930 - 1.026 \log M} / 2 \tag{6}$$

($M$ in kg), which we derived from **Stephens et al., 2019**. There were no specific invertivore or taxonomically specific equations to estimate $D$; however, we determined that the equation for the fitted 97.5 percentile in mammalian carnivores:

$$D = 10^{1.91 - 1.02 \log_{10} M} / 2 \tag{7}$$

**Stephens et al., 2019** provided is a reasonable $D$ for female *Tachyglossus* = 9.9 km$^{-2}$. This is comparable to echidna densities measured for Kangaroo Island (4.4 females km$^{-2}$) (**Rismiller and McKelvey, 2000**) and Tasmania (8.4 females km$^{-2}$) (**Nicol and Andersen, 2007**). We therefore also used *Equation 7* to predict $D$ for *Megalibgwilia*. For a detailed description of the distribution and trends of equilibrium densities, population sizes, and biomasses across the modelled species, see Appendix 3.

We estimated the maximum age ($\omega$) of each species according to:

$$\omega = 10^{0.89 + 0.13 \log_{10} M} \tag{8}$$

for non-volant birds and mammals ($M$ in g) (**Healy et al., 2014**), or

$$\omega = 7.02 M^{0.174} \tag{9}$$

for *Alectura* ($M$ in g) (**de Magalhães et al., 2007**), the latter of which produces a $\omega$ that closely matches the maximum longevity of 25 years estimated for the similar-sized megapode *Leipoa ocellata* (malleefowl) (**Bode and Brennan, 2011**).

For other species, we made species- or group-specific adjustments to the estimates of $\omega$: for *Vombatus* (*McIlroy, 2008*), we set $\omega = 26$; the disparity between this and the $\omega$ derived from the allometric prediction (*Equation 7* gives $\omega = 29$) means we adjusted $\omega'=26/29\omega$ for all vombatiform herbivores. Similarly for the macropodiformes, we scaled the predicted $\omega$ according to the degree of overprediction of the parameter from the equation for *Osphranter rufus* (for the latter, the equation predicted $\omega = 30$, but in reality it is closer to 13) (*Jonzén et al., 2010*), meaning we adjusted $\omega'=13/30\omega$ for all macropodiformes except *Notamacropus*; for *Notamacropus rufogriseus*, we took the mean maximum age (*Grzimek, 1990*) of 16. For *Dromaius* (*Atlas of Living Australia, 2020*),we set $\omega = 17$; for *Thylacoleo*, we set $\omega = 17$ to match female lion *Panthera leo* longevity (*Packer et al., 1998*); for *Thylacinus* (*Corbett, 1995*; *Prowse et al., 2014*), we set $\omega = 10$, for *Sarcophilus* (*Bradshaw and Brook, 2005*) $\omega = 5$, and *Dasyurus* $\omega = 4$ given the catastrophic mortality at maximum lifespan characteristic of dasyurids (*Cockburn, 1997*; *Holz and Little, 1995*; *Oakwood et al., 2001*). In the case of *Dasyurus* (both *D. maculatus* and *D. hallucatus*), most females die in their third year, although some can persist into the fourth year (*Cremona et al., 2017*; *Glen, 2008*; *Glen and Dickman, 2013*; *Moro et al., 2019*) and maximum longevity in captivity can be up to 5 years (*Way, 1988*). For *Tachyglossus*, $\omega$ is extremely high compared to similar-sized marsupials or placentals: up to 50 years in captivity, and possibly 45 years in the wild (*Nicol and Andersen, 2007*); we set the latter value of $\omega = 45$ to be conservative. For *Megalibgwilia*, we assumed that the underestimation for *Tachyglossus* according to *Equation 8* would also apply, so here we set *Megalibgwilia* $\omega = (45/23)26 = 51$ years.

We estimated fecundity ($F$; mean number of female neonates produced per year and per breeding female) for mammals (*Allainé et al., 1987*) as:

$$F = e^{2.719-0.211\log M}/2 \tag{10}$$

dividing by two for daughters only ($M$ in g). Although we used well-establish allometric relationships to derive our input parameters, most of these relationships are based on placental species. It is accepted that average life-history traits differ between similar-sized marsupials and placentals (*Fisher et al., 2001*), and we therefore estimated a correction factor for $F$ and age at first breeding ($\alpha$) (see *Equation 12*) for both the vombatiform and macropodiform herbivore groups separately (see *Appendix 5 figures 1–3* for approach) using demographic data describing marsupial life histories (*Fisher et al., 2001*).

For *Vombatus* (*McIlroy, 1996*), we set $F = 0.25$ given a litter size = 1, an inter-birth interval of 2 years, and an assumed sex ratio of 1:1, from which we derived $F$ for the extinct vombatiform herbivores (*Appendix 5—figure 2*). For *Notamacropus rufogriseus*, we set the average annual number of offspring = 1 multiplied by a 2.8% twinning rate (*Catt, 1977*), 1.3 based on an interbirth interval of 286 days, and an assumed 1:1 sex ratio. For *Genyornis*, we applied the following equation:

$$F = e^{2.35+0.17\log M}/2 \tag{11}$$

($M$ in g) (*Allainé et al., 1987*). For *Dromaius* (*Sales, 2007*), we used the average of 6.7 eggs clutch$^{-1}$ and 3.4 clutches year$^{-1}$, an assumed sex ratio of 1:1, and nest success (0.406) and hatching probabilities (0.419) for ostriches (*Kennou Sebei et al., 2009*). For *Alectura*, we used the annual mean of 16.6 eggs pair$^{-1}$ for *Leipoa ocellata* (*Bode and Brennan, 2011*), a hatching success of 0.866 for *Alectura lathami* (*Jones, 1988*), and an assumed 1:1 sex ratio. For *Thylacoleo*, we applied the average litter size of 1 for large vombatiforms, and a 1:1 sex ratio; for *Thylacinus* and *Sarcophilus*, we applied the values of 3.42 progeny litter$^{-1}$ and the proportion (*Lachish et al., 2009*; *Prowse et al., 2014*) of adults reproducing year$^{-1}$ (0.91), and a 1:1 sex ratio. The allometric prediction (*Equation 10*) nearly matched the product of mean litter size (4.9) (*Glen, 2008*) and proportion females breeding (0.643) (*Glen and Dickman, 2013*) for *Dasyurus*, so we used the former. For *Tachyglossus* (*Nicol and Morrow, 2012*), we used the production of 1 egg breeding event (divided by two for daughters) multiplied by the probability of breeding = 0.55. For *Megalibgwilia*, we assumed that the overestimation for *Tachyglossus* according to *Equation 10* would also apply, so here we set *Megalibgwilia* $F = (0.275/0.659)0.532 = 0.222$.

To estimate the age at first breeding ($\alpha$), we used the following relationship for mammals (*de Magalhães et al., 2007*):

$$\alpha = e^{-1.34+2.14\log M} \tag{12}$$

and

$$\alpha = 0.214M^{0.303} \tag{13}$$

for birds (*de Magalhães et al., 2007*) (*M* in g). For the macropodiforms, Equation 12 appeared to overestimate $\alpha$ by ~ 20% (see *Appendix 5—figure 2*), so we adjusted the extinct macropodiforms accordingly. For the vombatiform herbivores, Equation 12 performed more according to expectation. For example, the 1800–2000 kg female white rhinoceros (*Ceratotherium simum*) has $\alpha$ = 6–7 years (*Wilson and Mittermeier, 2001*), which is similar to the 7 years predicted by equation 12 for the 2786 kg *Diprotodon optatum* (but < $\alpha$ = 10–12 years for the > 6000 kg female African savanna elephant *Loxodonta africana*) (*Asier, 2016*). We also made species-specific adjustments to $\alpha$ for the extant species (or recently extinct in the case of *Thylacinus*). In the case of *Vombatus* (*Roger et al., 2011*), we set $\alpha$ = 2, $\alpha$ = 2 for *Osphranter rufus* (*Jonzén et al., 2010*), $\alpha$ = 1 for *Notamacropus rufogriseus* (*Catt, 1977*), $\alpha$ = 3 for *Dromaius* (*Patodkar et al., 2009*), $\alpha$ = 2 for *Alectura* based on data from *Leipoa ocellata* (*Bode and Brennan, 2011*; *Frith, 1959*), and $\alpha$ = 1 for *Thylacinus* (*Lachish et al., 2009*; *Prowse et al., 2014*), *Sarcophilus* (*Bradshaw and Brook, 2005*), and *Dasyurus* (*Glen and Dickman, 2013*). For *Thylacinus* and *Sarcophilus*, although $\alpha$ = 1, only a small proportion of females breed at this age (see below), so for most females $\alpha$ is in fact 2. But for *Dasyurus*, some females can become sexually mature and breed at < 12 months old (*Moro et al., 2019*), so we incorporated a modest capacity to reproduce in the year prior to age 1 (40% of total fertility). For *Tachyglosssus*, we set $\alpha$ = 3 based on evidence that echidnas take 3–5 years to reach adult mass (*Nicol and Andersen, 2007*), and only adults are observed to breed (*Rismiller and McKelvey, 2000*); as such, we set *m* = 0.5*F* in year 3, 0.75*F* in year 4, and *m* = *F* thereafter. As we did for $\omega$ and *F*, we estimated the bias between the allometrically predicted and measured $\alpha$ (equation 12) for *Tachyglossus*, and applied this to *Megalibgwilia*; however, rounding to the nearest year also means $\alpha$ = 3 for *Meglibgwilia*. The $\log_{10}$ of the resulting $\alpha$ among species predicted their respective $\log_{10}$ $r_m$ well ($R^2$ = 0.73) (*Appendix 5—figure 4*), with the fitted parameters similar to theoretical expectation (*Hone et al., 2010*), and thus, supporting our estimates of $r_m$ as realistic.

To estimate age-specific fertilities ($m_x$) from *F* and $\alpha$, we fit a logistic power function of the general form:

$$m_x = \frac{a}{1 + \left(\frac{x}{b}\right)^c} \tag{14}$$

where *x* = age in years, and *a*, *b*, *c* are constants estimated for each species, to a vector composed of ($\alpha - 1$) values at 0*F*, $\lfloor \alpha/2 \rfloor$ values at 0.75*F*, and for the remaining ages up to $\omega$, the full value of *F*. This produced a continuous increase in $m_x$ up to maximum rather than a less-realistic stepped series. For *Sarcophilus*, we instead used the parameters from an existing devil model (*Bradshaw and Brook, 2005*) to populate the $m_x$ vector.

To estimate realistic survival schedules, we first used the allometric prediction of adult survival ($s_{ad}$) as:

$$S_{ad} = e^{-e^{-0.5-0.25\log M}} \tag{15}$$

for mammals, and:

$$S_{ad} = e^{-e^{-1.78-0.21\log M}} \tag{16}$$

for birds, where *M* = body mass (g) (*McCarthy et al., 2008*). For *Tachyglossus*, we used the mean $S_{ad}$ = 0.96 based on upper and lower estimates of mortality for tagged individuals over 15 years in Tasmania (*Nicol and Morrow, 2012*), and applying the allometric-bias correction for *Megalibgwilia* as for $\omega$, *F*, and $\alpha$ as described above. We then applied the Siler hazard model (*Gurven and Kaplan, 2007*) to estimate the age- (*x*-) specific proportion of surviving individuals ($l_x$); this combines survival schedules for immature, mature, and senescent individuals within the population:

$$l_x = e^{\left(\frac{-a_1}{b_1}\right)\left(1-e^{-b_1 x}\right)} e^{-a_2 x} e^{\left(\frac{a_3}{b_3}\right)\left(1-e^{b_3 x}\right)} \qquad (17)$$

where $a_1$ = initial immature mortality, $b_1$ = rate of mortality decline in immatures, $a_2$ = the age-independent mortality due to environmental variation, $a_3$ = initial adult mortality, and $b_3$ = the rate of mortality increase (senescence). From $l_x$, age-specific survival can be estimated as:

$$S_x = 1 - \frac{(l_x - l_{x+1})}{l_x} \qquad (18)$$

We estimated the component parameters starting with $1 - S_{ad}$ for $a_1$ and $a_2$, adjusting the other parameters in turn to produce a dominant eigen value ($\lambda_1$) from the transition matrix containing $S_x$ (see Materials and methods) such that $\log_e \lambda_1 \approx r_m$. However, in many cases, marsupial and monotreme life histories were incapable of reproducing predicted $r_m$ (see *Appendix 2—table 1* below), although we attempted to maximize $\log_e \lambda_1$ wherever possible. This appears to be biologically justified given the slower life histories of vombatiforms and monotremes in particular compared to macropodiforms. We also generally favoured a more pronounced senescence component of $l_x$ in the longer-lived species given evidence for survival senescence in long-lived mammals (*Turbill and Ruf, 2010*). For *Sarcophilus*, we instead used the parameters from an existing devil model (*Bradshaw and Brook, 2005*) to populate the $S_x$ vector.

**Appendix 2—table 1.** Predicted demographic values for each species (equations provided in Materials and methods and this appendix).

$M$ = mass, $r_m$ = maximum rate of instaneous exponential population growth predicted allometrically, $r'_m$ = realised $r_m$ predicted from the constructed matrix (see text), $\omega$ = longevity, $F$ = fertility (daughters per breeding female per year), $\alpha$ = age at first reproduction (primiparity), $S_{ad}$ = yearly adult survival, $G$ = generation length. [†]extinct; [♀]extant. See *Appendix 2—table 2* for rank correlations among demographic values across species.

| Species | $M$ (kg) | $r_m$ | $r'_m$ | $D$ (km$^{-2}$) | $\omega$ (yrs) | $F$ ($n_\female$yr$^{-1}\female^{-1}$) | $\alpha$ (yrs) | $S_{ad}$ (yr$^{-1}$) | $G$ (yrs) |
|---|---|---|---|---|---|---|---|---|---|
| **vombatiform herbivores** | | | | | | | | | |
| *Diprotodon*[†] | 2786 | 0.100 | 0.061 | 0.134 | 48 | 0.1311 | 7 | 0.985 | 18.1 |
| *Palorchestes*[†] | 1000 | 0.131 | 0.077 | 0.285 | 42 | 0.1705 | 6 | 0.981 | 15.1 |
| *Zygomaturus*[†] | 500 | 0.157 | 0.095 | 0.476 | 39 | 0.2038 | 5 | 0.977 | 13.2 |
| *Phascolonus*[†] | 200 | 0.200 | 0.121 | 0.938 | 34 | 0.2586 | 4 | 0.972 | 10.7 |
| *Vombatus*[♀] | 25 | 0.345 | 0.119 | 4.370 | 26 | 0.2500 | 2 | 0.953 | 10.0 |
| **macropodiform herbivores** | | | | | | | | | |
| *Proctoptodon*[†] | 250 | 0.189 | 0.188 | 0.795 | 17 | 0.524 | 3 | 0.973 | 8.3 |
| *Sthenurus*[†] | 150 | 0.216 | 0.215 | 1.161 | 17 | 0.617 | 3 | 0.970 | 8.1 |
| *Protemnodon*[†] | 130 | 0.224 | 0.224 | 1.290 | 16 | 0.646 | 3 | 0.969 | 7.8 |
| *Simosthenurus*[†] | 120 | 0.229 | 0.226 | 1.369 | 16 | 0.663 | 3 | 0.968 | 7.8 |
| *Metasthenurus*[†] | 55 | 0.281 | 0.280 | 2.438 | 14 | 0.858 | 2 | 0.961 | 6.0 |
| *Osphranter*[♀] | 25 | 0.345 | 0.343 | 4.370 | 13 | 0.750 | 2 | 0.953 | 5.5 |
| *Notamacropus*[♀] | 14 | 0.402 | 0.351 | 6.712 | 16 | 0.668 | 1 | 0.993 | 6.3 |
| **large birds** | | | | | | | | | |
| *Genyornis*[†] | 200 | 0.041 | 0.041 | 0.101 | 38 | 0.658 | 9 | 0.987 | 20.0 |
| *Dromaius*[♀] | 55 | 0.100 | 0.100 | 0.290 | 17 | 1.938 | 3 | 0.983 | 5.9 |
| *Alectura*[♀] | 2.2 | 0.176 | 0.175 | 3.633 | 27 | 7.188 | 2 | 0.967 | 6.8 |
| **carnivores** | | | | | | | | | |
| *Thylacoleo*[†] | 110 | 0.234 | 0.201 | 0.028 | 14 | 0.500 | 4 | 0.967 | 9.1 |

*Continued on next page*

*Appendix 2—table 1 continued*

| Species | M (kg) | $r_m$ | $r'_m$ | D (km$^{-2}$) | ω (yrs) | F (n$_♀$yr$^{-1}$♀$^{-1}$) | α (yrs) | S$_{ad}$ (yr$^{-1}$) | G (yrs) |
|---|---|---|---|---|---|---|---|---|---|
| *Thylacinus*[†] | 20 | 0.366 | 0.368 | 0.159 | 10 | 1.556 | 1 | 0.950 | 5.2 |
| *Sarcophilus*[†,♀] | 6.1 | 0.500 | 0.094 | 0.539 | 5 | 1.205 | 1 | 0.820 | 3.1 |
| *Dasyurus*[♀] | 2.0 | 0.701 | 0.644 | 2.023 | 4 | 1.582 | 1 | 0.910 | 2.3 |
| **monotremes** | | | | | | | | | |
| *Megalibgwilia*[†] | 11.0 | 0.307 | 0.107 | 3.522 | 51 | 0.222 | 3 | 0.977 | 16.4 |
| *Tachyglossus*[♀] | 4.0 | 0.400 | 0.112 | 9.883 | 45 | 0.275 | 3 | 0.950 | 14.1 |

*Sarcophilus harrisii* is extinct in mainland Australia, but extant in the island state of Tasmania.

*Thylacinus* could also be treated like *Sarcophilus* in that *Thylacinus* survived in Tasmania until historical times (1930s).

In the case of the vombatiform and macropodiform herbivores, ω shown in the table is in fact the downscaled ω′ calculated for each group (see below). Likewise, both allometric predictions of F and α are corrected for these groups (see Supplementary Information Appendix 5).

**Appendix 2—table 2.** Kendall's rank correlation ($\tau$) among demographic values given in *Appendix 2—table 1* across species.

$M$ = mass, $r_m$ = maximum rate of instaneous exponential population growth predicted allometrically, $r'_m$ = realised $r_m$ predicted from the constructed matrix (see text), ω = longevity, F = fertility (daughters per breeding female per year), α = age at first reproduction (primiparity), $S_{ad}$ = yearly adult survival, G = generation length.

| | M (kg) | $r_m$ | rm′ | D (km$^{-2}$) | ω (yrs) | F (n$_♀$yr$^{-1}$♀$^{-1}$) | α (yrs) | S$_{ad}$ (yr$^{-1}$) |
|---|---|---|---|---|---|---|---|---|
| $r_m$ | −0.684 | | | | | | | |
| $r_m$′ | −0.350 | 0.507 | | | | | | |
| D (km$^{-2}$) | −0.476 | 0.499 | 0.358 | | | | | |
| ω (yrs) | 0.390 | −0.511 | −0.605 | −0.068 | | | | |
| F (n$_♀$yr$^{-1}$♀$^{-1}$) | −0.484 | 0.278 | 0.486 | 0.148 | −0.567 | | | |
| α (yrs) | 0.671 | −0.685 | −0.583 | −0.459 | 0.556 | −0.562 | | |
| S$_{ad}$ (yr$^{-1}$) | 0.552 | −0.681 | −0.394 | −0.328 | 0.484 | −0.327 | 0.572 | |
| G (yrs) | 0.471 | −0.446 | −0.616 | −0.201 | 0.762 | −0.711 | 0.730 | 0.487 |

## Appendix 3

### Allometric predictions of equilibrium population density, total population size, and biomass

The allometric predictions of stable population size ($N_{stable}$) for each species in the 500 km × 500 km study area showed the largest populations for some of the smallest, extant species (e.g., $N_{stable}$ > 600,000,000 for *Vombatus*, *Osphranter*, *Notamacropus*, *Alectura*, *Dasyurus*, *Tachyglossus*) (*Appendix 3—figure 1a*). There was a clear separation in the allometric predictions of $N_{stable}$ among the species in each group (*Appendix 3—figure 1b*). When expressed as total biomass across the study area, the four carnivores had approximately equal biomasses (~$10^6$ kg), as did the macropodiformes and monotremes (*Appendix 3—figure 1c*). For the large birds and herbivore vombatiformes, biomass increased with body mass (*Appendix 3—figure 1c*).

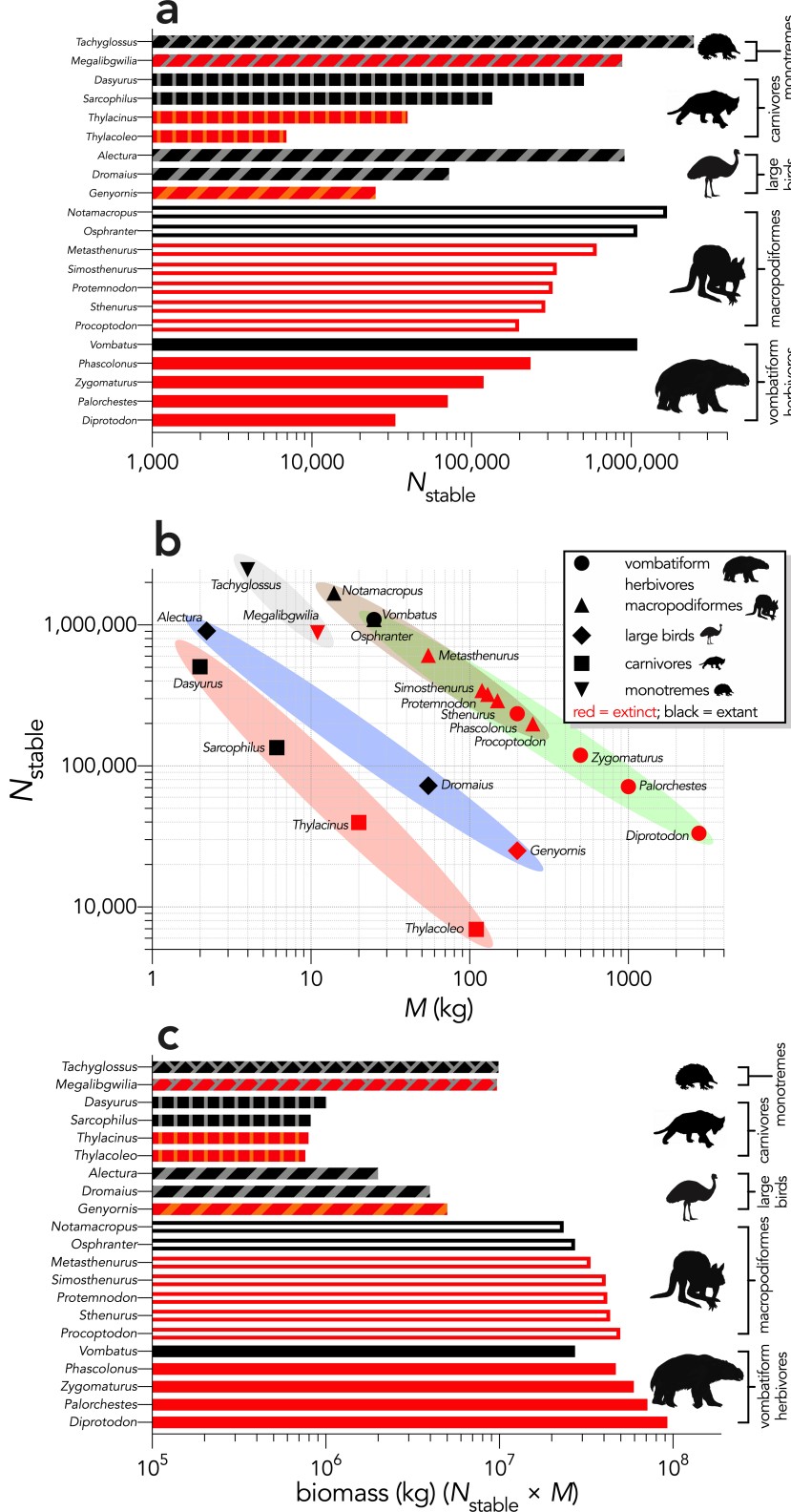

**Appendix 3—figure 1.** Mass-predicted abundance and biomass for the 21 modelled species. (**a**) Stable population sizes ($N_{stable}$) for each modelled species predicted from allometric estimates of population density for a 500 km × 500 km (250,000 km²) landscape (outline-only bars = macropodiformes; solid bars = vombatiforms; angled crosshatching = birds; vertical crosshatching =

*Appendix 3—figure 1 continued on next page*

*Appendix 3—figure 1 continued*

carnivores; brick crosshatching = monotremes); (**b**) $N_{stable}$ plotted against body mass ($M$, in kg), showing the allometric scaling separating the vombatiform herbivores (green)/macropodiformes (brown), flightless birds (blue), carnivores (red), and monotremes (grey); (**c**) predicted landscape biomass ($N_{stable} \times M$) for each species (outline-only bars = macropodiformes; solid bars = vombatiforms; angled crosshatching = birds; vertical crosshatching = carnivores; brick crosshatching = monotremes).

Here, we have depicted *Sarcophilus* as 'extant', even though it went extinct on the mainland >3000 years ago.

## Appendix 4

### Compensatory density feedback

To avoid an exponentially increasing population without limit generated by a transition matrix optimized to produce values as close to $r_m$ as possible, we applied a theoretical compensatory density-feedback function. This procedure ensures that the long-term population dynamics were approximately stable by creating a second logistic function of the same form as $m_x$ to calculate a modifier ($S_{mod}$) of the $S_x$ vector according to total population size ($\Sigma n$):

$$S_{mod} = \frac{a}{1 + \left(\frac{\sum n}{b}\right)^c} \tag{19}$$

We adjusted the $a$, $b$, and $c$ constants for each species in turn so that a stochastic projection of the population remained stable on average for 40 generations ($40\lfloor G \rceil$), where:

$$G = \frac{\log\left(\left(\mathbf{v}^T\mathbf{M}\right)_1\right)}{\lambda_1} \tag{20}$$

and $(\mathbf{v}^T\mathbf{M})_1$ = the dominant eigen value of the reproductive matrix $\mathbf{R}$ derived from $\mathbf{M}$, and $\mathbf{v}$ = the left eigenvector (*Caswell, 2001*) of $\mathbf{M}$. Although arbitrary, we chose a $40\lfloor G \rceil$ projection time as a convention of population viability analysis to standardize across different life histories (*Brook et al., 2006*; *Traill et al., 2010*).

The projections were stochastic in that we $\beta$-resampled the $S_x$ vector assuming a 5% standard deviation of each $S_x$ and Gaussian-resampled the $m_x$ vector at each yearly time step to $40\lfloor G \rceil$. We also added a catastrophic die-off function to account for the probability of catastrophic mortality events ($C$) scaling to generation length among vertebrates (*Reed et al., 2003*):

$$C = \frac{p_C}{G} \tag{21}$$

where $p_C$ = probability of catastrophe (*Reed et al., 2003*) (set at 0.14). Once invoked at probability $C$, we applied a $\beta$-resampled proportion centred on 0.5 to the $\beta$-resampled $S_x$ vector to induce a ~ 50% mortality event for that year (*Bradshaw et al., 2013*), as we assumed that a catastrophic event is defined as "… any 1 yr peak-to-trough decline in estimated numbers of 50% or greater" (*Reed et al., 2003*). Finally, for each species we rejected the first $\lfloor G \rfloor$ years of the projection as a burn-in to allow the initial (deterministic) stable stage distribution to stabilize to the stochastic expression of stability under compensatory density feedback.

## Appendix 5

### Deriving marsupial correction factors for fecundity (*F*) and age at first breeding ($\alpha$)

Given that allometric predictions of various life-history traits in mammals are based primarily on data from extant placentals, we investigated the degree of potential bias in our estimates of longevity ($\omega$) fecundity, (*F*) and age at first breeding ($\alpha$) based on a comparison of theoretical and observed data for extant marsupials. We discuss the bias correction $\omega$ for in the main text for the vombatiform and macropodiform herbivores (see Appendix 2), so here we report how we derived group-specific corrections for *F* and $\alpha$.

### Fertility (*F*) correction

We first collected adult female mass, inter-birth interval ($I_b$), and age at first breeding data for twenty-three extant species in the database compiled by *Fisher et al., 2001*. We included all species for which data were listed in the genera: *Macropus* (*Osphranter* and *Notamacropus*), *Dorcopsis*, *Lagorchestes*, *Petrogale*, *Thylogale*, and *Wallabia*. We excluded the genus *Dendrolagus* (tree kangaroos) because they represent a distinct clade and differ ecologically from most other macropodids. We also excluded the tammar wallaby (*Macropus eugenii*) because it is a strongly seasonal-breeding species that potentially strong leverage on estimating the allometric slope.

To correct *F*, we first examined the relationship between $I_b$ and body mass (*M*) for these species (*Appendix 5—figure 1*):

We therefore concluded that there was sufficient evidence for an allometric relationship between the two variables for this group, which we used to project the degree to which *F* was overestimated by the allometric relationship (*Equation 10*) used to estimate *F* for the extinct macropodidiform herbivores. Using the intercept and slope estimated in the relationship shown in *Appendix 5—figure 1*, we predicted an inter-birth interval of 384 days for *Procoptodon*, 363 days for *Sthenurus*, 357 days for *Protemnodon*, 354 days for *Simosthenurus*, and 322 days for *Metasthenurus*. These changed the allometrically predicted *F* by −4.9%, +0.5%, +2.2%, +3.1%, and +13.3%, respectively (corrected *F* shown in *Appendix 2—table 1*).

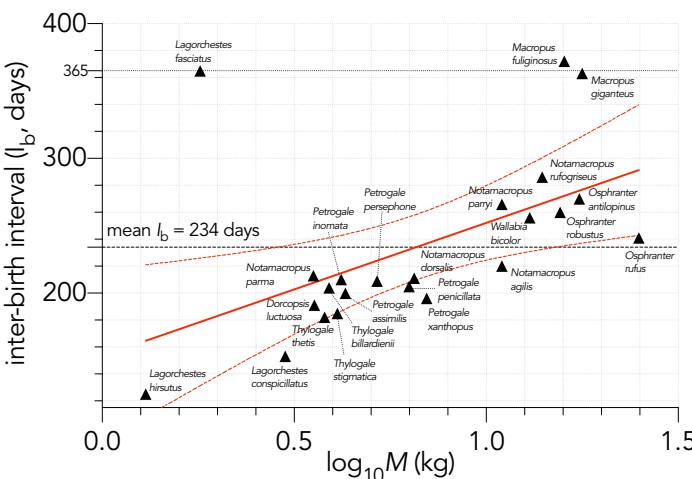

**Appendix 5—figure 1.** Relationship between the logarithm of adult female body mass (M, kg) and inter-birth interval ($I_b$, in days) for 23 extant macropodid herbivores (*Fisher et al., 2001*). The estimated parameters of the linear fit ($y \sim \alpha + \beta x$) are: $\alpha = 159.3 \pm 31.5$ days ($\pm$ SE) and $\beta = 93.6 \pm 36.4$, explaining 24.2% of the variation ($R^2_{\mathrm{adj}}$), with the information-theoretic evidence ratio (ER) of the slope *versus* intercept-only model = 11.0.

For the vombatiforms, there are only four extant phascolarctid (koala *Phascolarctos cinereus*) and vombatiform herbivores (common or bare-nosed wombat *Vombatus ursinus*, northern hairy-nosed wombat *Lasiorhinus krefftii*, and southern hairy-nosed wombat *L. latifrons*). There were not enough

species to estimate an allometric relationship that might predict the expected $I_b$ for extinct vombatiform herbivores, so instead we assumed that the extinct vombatiform herbivores we considered would scale allometrically relative to *Vombatus ursinus*, which has a measured $I_b$ of 730 days (*Fisher et al., 2001*). For this, we assumed the same slope as measured for the extant macropodiforms (*Appendix 5—figure 1*), and an intercept that aligned *Vombatus* with the relationship (*Appendix 5—figure 2*):

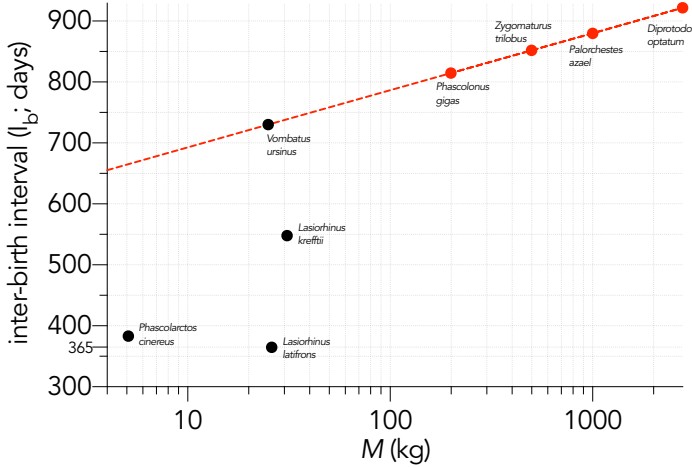

**Appendix 5—figure 2.** Relationship between the logarithm of adult female body mass (M, kg) and inter-birth interval ($I_b$, in days) for four large, extant phoscolarctid and vombatiform herbivores (black circles): koala *Phascolarctos cinereus*; common wombat *Vombatus ursinus*; northern hairy-nosed wombat *Lasiorhinus krefftii*; southern hairy-nosed wombat *L. latifrons*. Shown is the assumed relationship between $I_b$ and $\log_{10}M$ setting the slope to that estimated for the extant macropodiforms ($\beta$ = 93.6; *Appendix 5—figure 1*) and an intercept that aligned with *Vombatus* ($\alpha$ = 599 days) to estimate the inter-birth interval for the four extinct vombatiform herbivores (red circles) considered in this analysis.

## Age at first breeding (α) correction

Next, we estimated the bias in the predicted age at first breeding ($\alpha$) for the macropodiforms. A similar correction for the vombatiform herbivores was not warranted given that the allometric predictions were close to expectation for placentals of similar mass (see main text). We plotted $\alpha$ predicted from the allometric *Equation 12* against those observed from the marsupial database (*Fisher et al., 2001*) for the extant macropodiform species as described above for F (*Appendix 5—figure 3*):

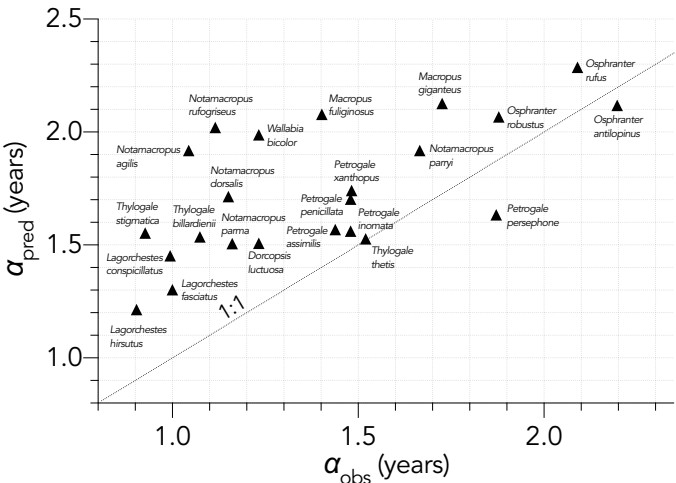

**Appendix 5—figure 3.** Relationship between the predicted age (years) at first breeding ($\alpha_{\text{pred}}$) and observed $\alpha$ ($\alpha_{\text{obs}}$) for 23 extant macropodid herbivores (*Fisher et al., 2001*). The allometric prediction over-estimated $\alpha$ by and average of ~20%. Also shown is the 1:1 line (dashed).

Calculating the average disparity between the predicted and observed $\alpha$ across species, the allometric prediction over-estimated $\alpha$ by 20% for the macropodiform herbivores. We therefore applied this correction factor to the estimated $\alpha$ for the extinct macropodiform species (corrected values in *Appendix 2—table 1*). The corrected $\alpha$ predicted maximum rate of population growth ($r_m$) approximately following theoretical expectation (see *Appendix 5—figure 4*).

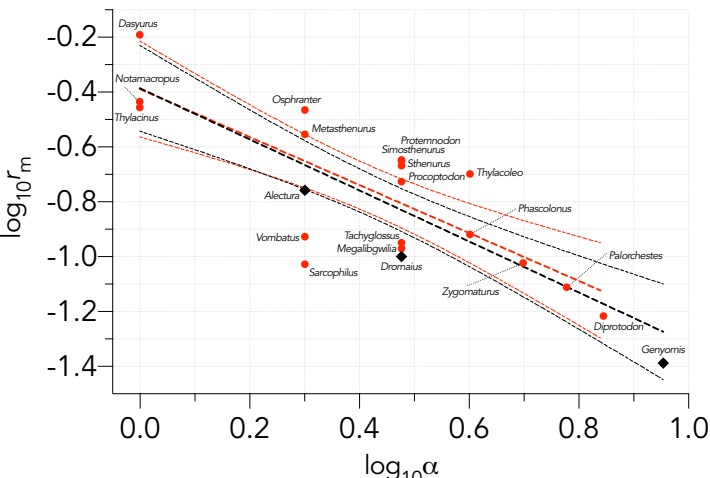

**Appendix 5—figure 4.** Negative relationship between the logarithm of the maximum rate of intrinsic population growth ($\log_{10} r_m$) and the logarithm of the age at primiparity ($\log_{10} \alpha$) for the 21 species examined. The estimated parameters of the linear fit ($y \sim \alpha + \beta x$) including all species (black lines) are: $\alpha$ = -0.388 ± 0.075 (± SE) and $\beta$ = -0.931 ± 0.146, and explaining 66.4% of the variation ($R^2_{\text{adj}}$), with the information-theoretic evidence ratio (ER) of the slope *versus* intercept-only model = $4.052 \times 10^4$. This relationship is similar to the theoretical expectation for the intercept = -0.15 and slope = -1.0 for mammals (*Hone et al., 2010*). Birds (*Alectura, Dromaius, Genyornis*; ◆) potentially fall outside this relationship, so just considering the remaining mammals (•), the parameters for the linear fit (red lines) become: $\alpha$ = -0.388 ± 0.083 (± SE), $\beta$ = -0.875 ± 0.170, $R^2_{\text{adj}}$ = 60.1%, and ER = $1.567 \times 10^3$.

## Appendix 6

### Quasi-extinction curves for each species in each of the six perturbation scenarios

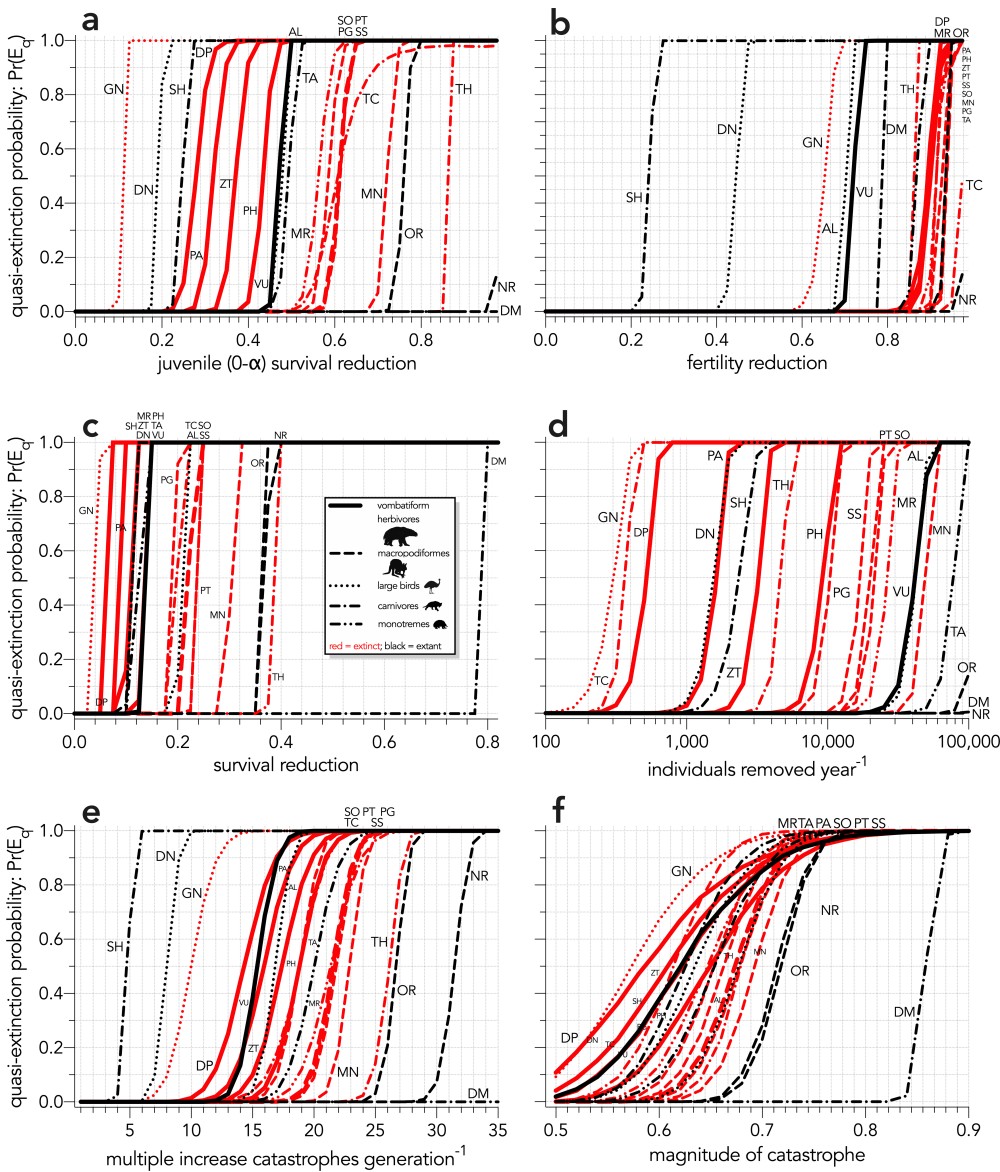

**Appendix 6—figure 1.** Increasing probabilities of quasi-extinction — quasi-extinction probability: Pr ($E_q$) — as a function of (a) decreasing juvenile survival (Scenario ↓$S_j$), (b) decreasing fertility (Scenario ↓$F$), (c) decreasing survival across all age classes (Scenario ↓$S$), (d) increasing number of individuals removed year$^{-1}$ (Scenario ↓ind), (e) increasing frequency of catastrophic die-offs per generation (Scenario ↑cat$_f$), and (f) increasing magnitude of catastrophic die-offs (Scenario ↑cat$_M$). Species notation: DP = *Diprotodon optatum*, PA = *Palorchestes azael*, ZT = *Zygomaturus trilobus*, PH = *Phascolonus gigas*, VU = *Vombatus ursinus* (vombatiform herbivores); PG = *Procoptodon goliah*, SS = *Sthenurus stirlingi*, PT = *Protemnodon anak*, SO = *Simosthenurus occidentalis*, MN = *Metasthenurus newtonae*, OR = *Osphranter rufus*, NR = *Notamacropus rufogriseus* (macropodiformes); GN = *Genyornis newtoni*, DN = *Dromaius novaehollandiae*, AL = *Alectura lathami* (large birds); TC = *Thylacoleo carnifex*, TH = *Thylacinus cynocephalus*, SH = *Sarcophilus harrisii*, DM = *Dasyurus maculatus* (carnivores); TA = *Tachyglossus aculeatus*, MR = *Megalibgwilia*

*Appendix 6—figure 1 continued on next page*

*Appendix 6—figure 1 continued*

*ramsayi* (monotreme invertivores). Here, we have depicted SH as 'extant', even though it went extinct on the mainland >3000 years ago.

## Appendix 7

### Comparing increasing mortality in juveniles and increasing offtake of juvenile individuals

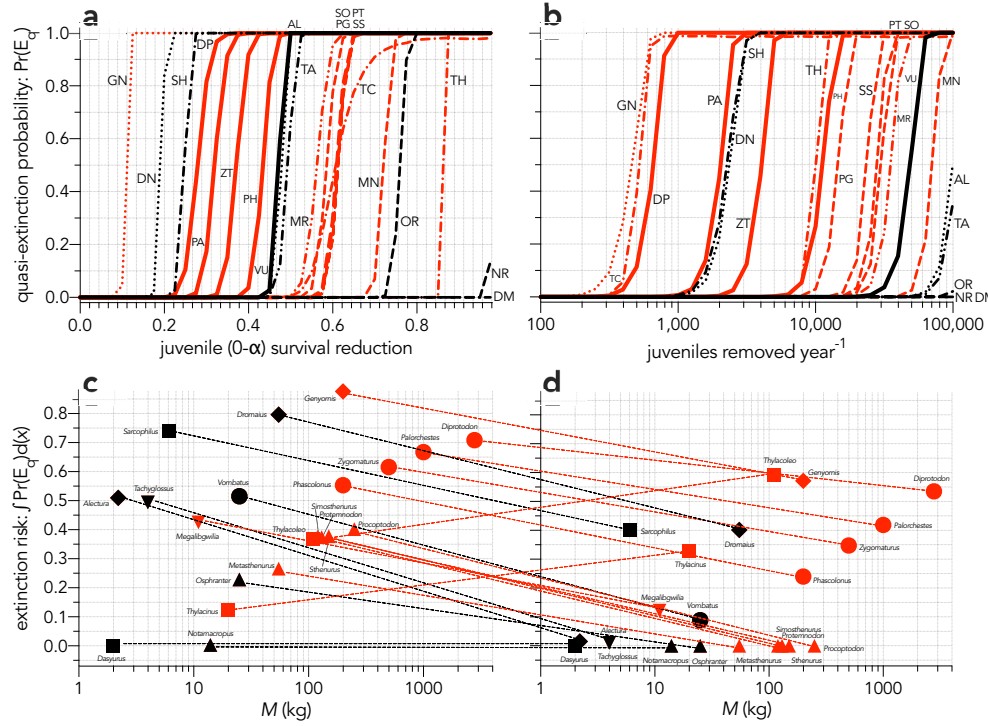

**Appendix 7—figure 1.** Increasing probabilities of quasi-extinction — quasi-extinction probability: Pr $(E_q)$ — as a function of (a) decreasing juvenile survival (Scenario $\downarrow S_j$), and (b) increasing number of juvenile individuals removed year$^{-1}$ (Scenario $\downarrow$ind$_j$). Also shown is the corresponding area under the quasi-extinction curve — extinction risk: $\int Pr(E_q)d(x)$ — as a function body mass for (c) increasing juvenile mortality and (d) increasing number of juvenile individuals removed year$^{-1}$. The dashed lines in c and d indicate the change in relative susceptibility between scenarios. Species notation: DP = *Diprotodon optatum*, PA = *Palorchestes azael*, ZT = *Zygomaturus trilobus*, PH = *Phascolonus gigas*, VU = *Vombatus ursinus* (vombatiform herbivores); PG = *Procoptodon goliah*, SS = *Sthenurus stirlingi*, PT = *Protemnodon anak*, SO = *Simosthenurus occidentalis*, MN = *Metasthenurus newtonae*, OR = *Osphranter rufus*, NR = *Notamacropus rufogriseus* (macropodiformes); GN = *Genyornis newtoni*, DN = *Dromaius novaehollandiae*, AL = *Alectura lathami* (large birds); TC = *Thylacoleo carnifex*, TH = *Thylacinus cynocephalus*, SH = *Sarcophilus harrisii*, DM = *Dasyurus maculatus* (carnivores); TA = *Tachyglossus aculeatus*, MR = *Megalibgwilia ramsayi* (monotreme invertivores). Red = extinct; black = extant. Here, we have depicted SH as 'extant', even though it went extinct on the mainland >3000 years ago.

## Appendix 8

### Sensitivity analysis of extinction dates using a jack-knife approach based on the Gaussian-Resampled, Inverse-Weighted McInerny (GRIWM) Signor-Lipps correction method

To test the sensitivity of our conclusion to uncertainty in the dates of final extinction for each taxon taken from *Saltré et al., 2016* and *White et al., 2018*, we adapted the Gaussian-resampled, inverse-weighted McInerny (GRIWM) approach (*Bradshaw et al., 2012*; *Saltré et al., 2015*) to derive new estimates and uncertainties of extinction time. Here, we jack-knifed the estimates 10,000 times for each taxon, calculating the median and 95% lower and upper confidence bounds. We applied this new approach to the following taxa: *Diprotodon*, *Genyornis*, *Megalibgwilia*, *Metasthenurus*, *Palorchestes*, *Phascolonus*, *Procoptodon*, *Protemnodon*, *Sismosthenurus*, *Sthenurus*, *Thylacoleo*, and *Zygomaturus* (but we kept the original dates for *Sarcophilus* and *Thylacinus*).

GRIWM does not depend on distributional assumptions given that it is a resampling approach (*Saltré et al., 2015*), even if it often provides wider uncertainty windows (especially for older dates) compared to other approaches. The main effect of this approach was to push back the extinction date for *Metasthenurus* (from a median of 50 ka to 89 ka), and push forward some of the dates within the other macropodiformes and vombatiforms (*Appendix 8 figures 1–3*). However, the net effect when we related these new estimates to any of the different risk metrics — mass (*Appendix 8—figure 1a*), generation length (*Appendix 8—figure 1b*), vital rate-specific extinction risk (*Appendix 8—figure 2*), or median susceptibility rank (*Appendix 8—figure 3*) — was to maintain our general conclusion of no relationship with the chronology.

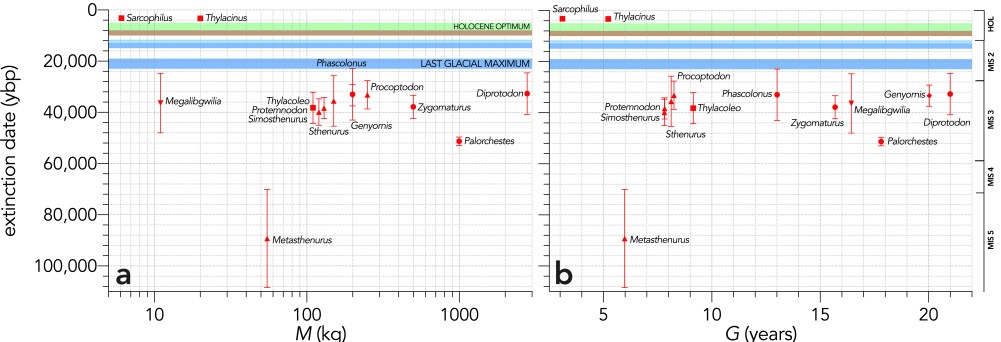

**Appendix 8—figure 1.** Relationship between estimated date of species extinction (across the entire continent) based on a jack-knifed GRIWM approach (*Bradshaw et al., 2012*; *Saltré et al., 2015*) and (**a**) body mass (kg) or (**b**) generation length (years) (Scenario LH). Extinction-timing windows are estimated based on the agreement among six different models that correct for the Signor-Lipps effect (described in Materials and methods) in chronologies of quality-rated (*Rodríguez-Rey et al., 2015*) fossil dates for the studied taxa described in *Peters et al., 2019*. Here, we have depicted *Sarcophilus* as 'extant', even though it went extinct on the mainland >3000 years ago. Also shown are the approximate major climate periods and transitions: Marine Isotope Stage 5 (MIS 5), MIS 4, MIS 3, MIS 2 (including the Last Glacial Maximum, Antarctic Cold Reversal, and Younger Dryas), and the Holocene (including the period of sea level flooding when Tasmania separated from the mainland, and the relatively warm, wet, and climatically stable Holocene optimum).

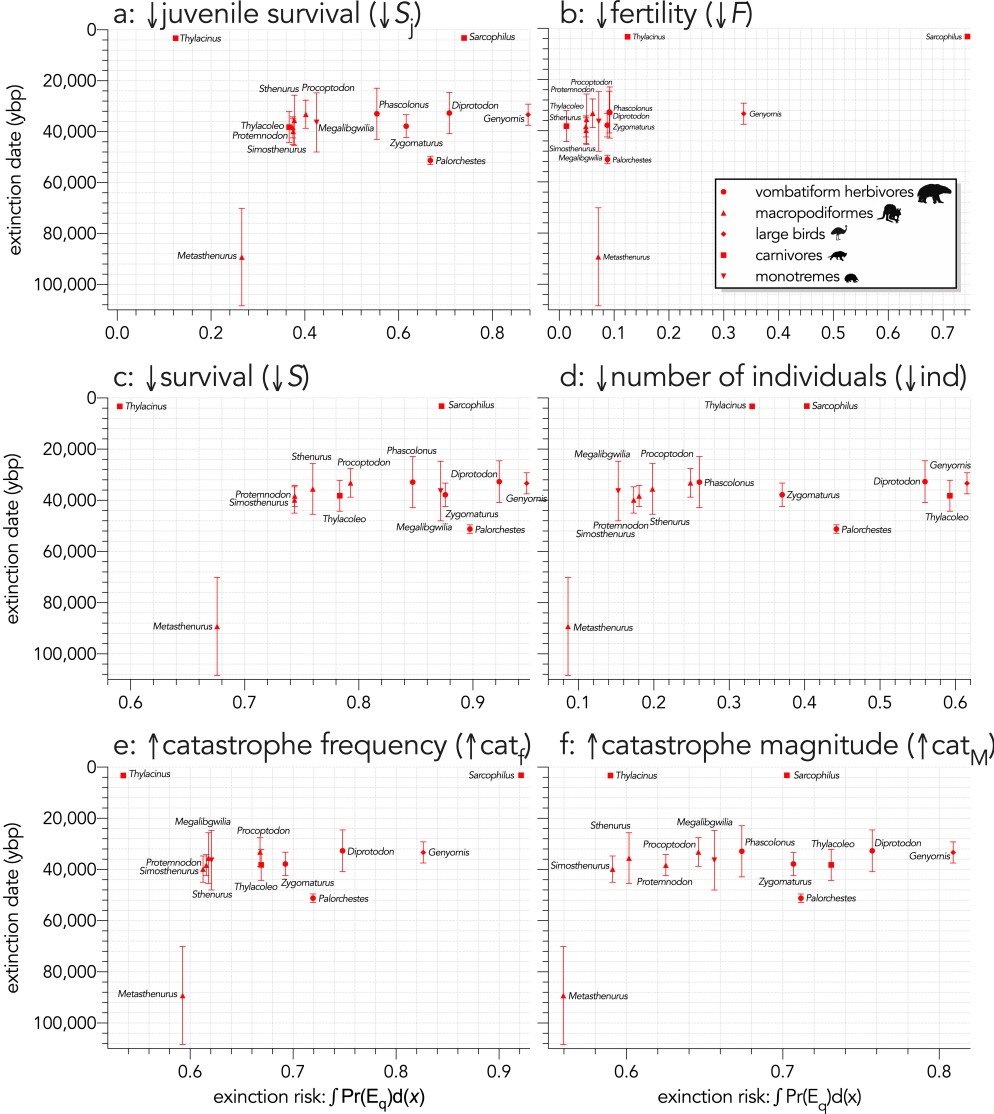

**Appendix 8—figure 2.** Relationship between estimated date of species extinction (across the entire mainland) based on a jack-knifed GRIWM approach (*Bradshaw et al., 2012*; *Saltré et al., 2015*) and area under the quasi-extinction curve (from Fig. S7) — extinction risk: ∫Pr(E_q)d(x) — for (**a**) (Scenario ↓$S_j$) decreasing juvenile survival, (**b**) (Scenario ↓$F$) decreasing fertility, (**c**) (Scenario ↓$S$) decreasing survival across all age classes, (**d**) (Scenario ↓ind) increasing number of individuals removed year$^{-1}$, (**e**) (Scenario ↑cat_f) increasing frequency of catastrophic die-offs per generation, and (**f**) (Scenario ↑cat_M) increasing magnitude of catastrophic die-offs. Extinction-timing windows are estimated based on the agreement among six different models that correct for the Signor-Lipps effect (described in Materials and methods) in chronologies of quality-rated (*Rodríguez-Rey et al., 2015*) fossil dates for the studied taxa described in *Peters et al., 2019*.

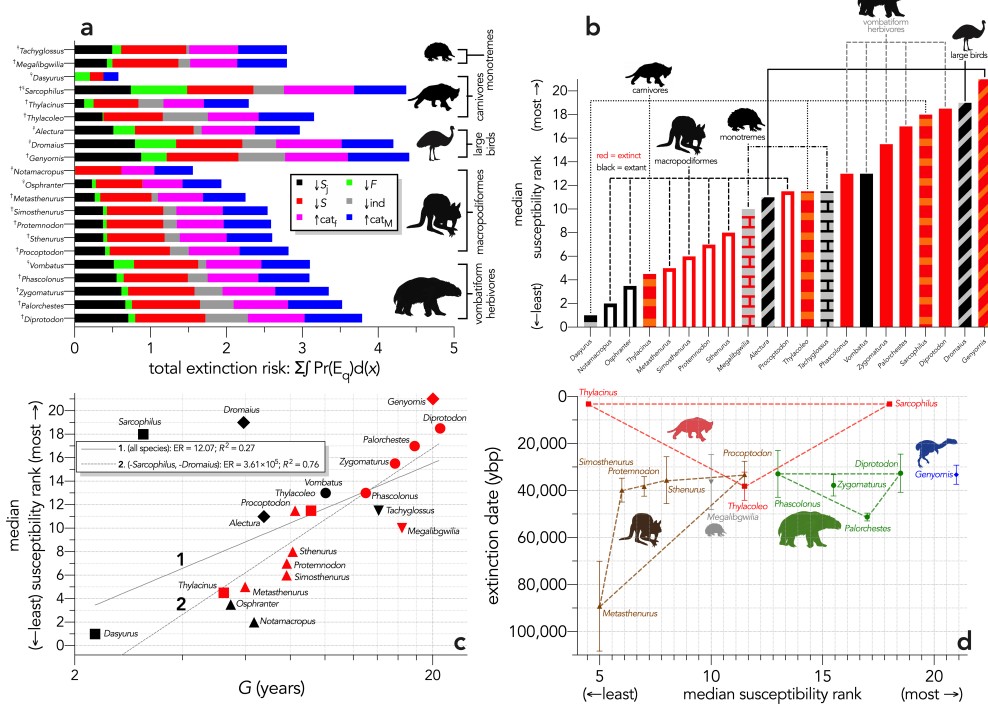

**Appendix 8—figure 3.** (**a**) Sum of the areas under the quasi-extinction curve for each of the six scenarios considered — total extinction risk: $\Sigma \int Pr(E_q)d(x)$ — for each of the 21 modelled species ([†]extinct; [♀]extant; scenario abbreviations: $\downarrow S_j$ = reducing juvenile survival; $\downarrow F$ = reducing fertility; $\downarrow S$ = reducing survival; ind = reducing number of individuals; $\uparrow cat_f$ = increasing frequency of catastrophe; $\uparrow cat_M$ = increasing magnitude of catastrophe); (**b**) median susceptibility rank across the six scenarios considered (where higher ranks = higher susceptibility to extinction) for each species (red = extinct; black = extant; outline-only bars = macropodiformes; solid bars = vombatiforms; angled crosshatching = birds; vertical crosshatching = carnivores; brick crosshatching = monotremes); (**c**) median susceptibility rank as a function of $\log_{10}$ generation length ($G$, kg) — there was a weak correlation including all species (solid grey line 1), but a strong relationship after removing *Sarcophilus* and *Dromaius* (dashed grey line 2) (information-theoretic evidence ratio [ER] and variance explained [$R^2$] shown for each); (**d**) estimated date of species extinction (across entire continent) as a function of median susceptibility rank; taxonomic/functional groupings are indicated by colored symbols and convex hulls (macropodids: brown; monotremes: grey; vombatiforms: green; birds: blue; carnivores: red). Extinction-timing windows are on a jack-knifed GRIWM approach (*Bradshaw et al., 2012*; *Saltré et al., 2015*) that corrects for the Signor-Lipps effect in chronologies of quality-rated (*Rodríguez-Rey et al., 2015*) fossil dates for the studied taxa described in *Peters et al., 2019*.

## Appendix 9

### Comparing demographic susceptibility to climate variation

Failing to observe any relationship between overall demographic susceptibility and the extinction chronology for Sahul, we might alternatively expect species' demographic susceptibility to align with increasing environmental stress expressed as hotter, drier climates ( *Saltré et al., 2019*). We therefore hypothesized that the most extreme (hottest/driest) climates of the past would eventually drive the most-resilient species to extinction, which would manifest as a negative relationship between demographic susceptibility and warming/drying conditions (i.e., only when conditions became bad enough did the least-susceptible species succumb).

To test this hypothesis, we compiled climate indices hindcasted for the estimated extinction windows for the species we considered here. To this end, we acquired four hindcasted, continentally averaged climate variables from the intermediate-complexity, three-dimensional, Earth-system model known as LOVECLIM (*Goosse et al., 2010*; *Timmermann and Friedrich, 2016*). LOVECLIM hindcasts various climatic conditions by incorporating representations of the atmosphere, ocean and sea ice, land surface (including a vegetation submodel), ice sheets, and the carbon cycle. These variables were — mean annual temperature (°C), mean annual precipitation (mm), net primary production (kg C m$^{-2}$ year$^{-1}$), and fraction of the landscape designated as 'desert' — all expressed as anomalies of their respective average values calculated relative to 120 ka (i.e., a time when all species we considered were extant). We downscaled the original spatial resolution of LOVECLIM (5.625°×5.625°) to an output resolution of 1°×1° using bilinear interpolation because it retains the integrity and limitations of the original model output data.

We then calculated the information-theoretic evidence ratios (ER) for all relationships between the mean value of the climate variable across the entirely of Sahul and the sum of the extinction integrals across scenarios as the Akaike's information criterion (AIC) of the slope model: $y = \alpha + \beta x$ divided by the AIC of the intercept-only model: $y = \alpha$ (i.e., $ER_{mean} = AIC_{slope}/AIC_{intercept}$). To incorporate full uncertainty in the climate variables ($y$), we developed a randomization test where we uniformly resampled the $y$ values between $y_{min}$ and $y_{max}$, estimating the residual sum of squares of the resampled values at each iteration compared to a randomized order of these residuals. We then calculated the probability ($p_u$) of producing a randomly generated relationship between the climate variable and the sum of the extinction integrals as the number of iterations when the randomized order produced a residual sum of squares $\leq$ the residual sum of squares of the resampled (ordered) climate variables divided by the total number of iterations (10,000).

When we plotted the four climate variables against the sum of the quasi-extinction integrals across scenarios, there was evidence for a weak, negative relationship between mean annual precipitation anomaly and relative extinction susceptibility ($ER_{mean} = 9.6$; *Appendix 9—figure 1b*), and a weak, positive relationship with desert-fraction anomaly ($ER_{mean} = 4.2$; Fig. *Appendix 9—figure 1d*). There was no evidence for a relationship between the means for temperature (*Appendix 9—figure 1a*) and net primary production (*Appendix 9—figure 1c*) anomalies and quasi-extinction integrals. The relationship with desert fraction supports the hypothesis that a drier (more desert-like) environment might have been related to extinction susceptibility. However, when we took full uncertainty of the climate variables into account in a randomization least-squares regression, none of the relationships could not be differentiated from a random process ($p_u > 0.17$; *Appendix 9—figure 1*).

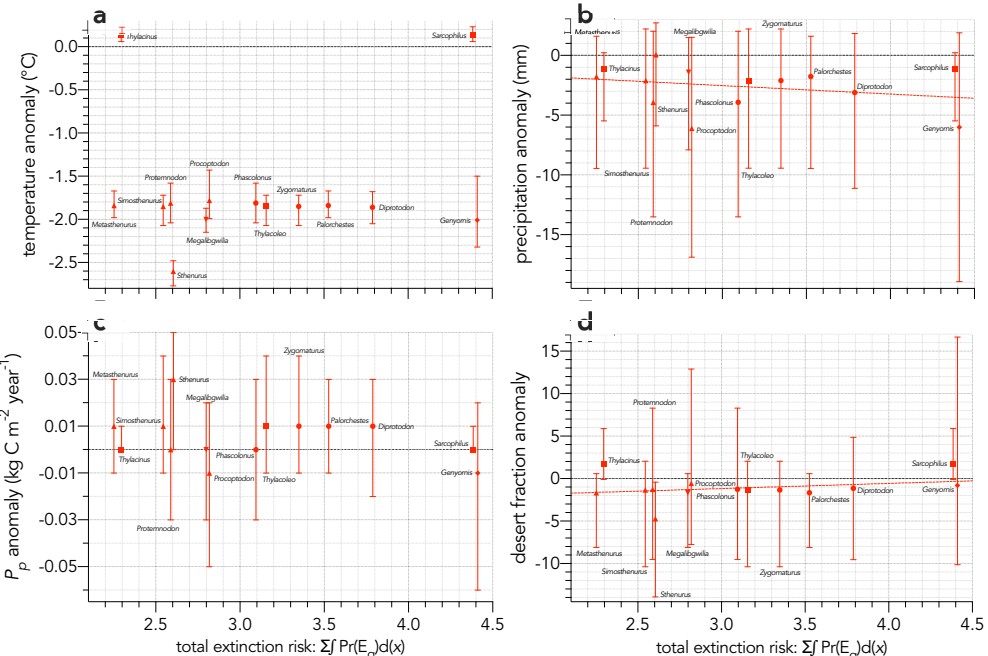

**Appendix 9—figure 1.** Sum of the areas under the quasi-extinction curve over the six scenarios considered — total extinction risk: $\Sigma \int Pr(E_q)d(x)$ — for each of the 13 extinct (mainland only) modelled species relative to (**a**) mean annual temperature anomaly (°C): information-theoretic evidence ratio of the slope model relative to the intercept-only model ($ER_{mean}$) = 0.75 for the mean climate values; probability of a non-random slope relationship incorporating full uncertainty in the climate variable $p_u$ = 0.603. (**b**) mean annual precipitation anomaly (mm): $ER_{mean}$ = 9.6; $p_u$ = 0.461. (**c**) net primary production anomaly (kg C m$^{-2}$ year$^{-1}$): $ER_{mean}$ <0.01; $p_u$ = 0.411. (**d**) desert fraction anomaly: $ER_{mean}$ = 4.2; $p_u$ = 0.425. The dashed red line in panels a, b, and d indicate evidence for a slope model versus the intercept-only model for these variables ($ER_{mean}$ >2). Error bars indicate ±1 standard deviation.

The lack of evidence for a relationship between extinction susceptibility and warming/drying conditions for the mean climate conditions across the continent contrasts with recent evidence that water availability potentially exacerbated mortality from novel human hunting (*Saltré et al., 2019*). However, there is too much uncertainty in the climate hindcasts to test this hypothesis definitively. Another weakness of this approach is that we were obliged to take continental-scale averages of average climate conditions, which obviously ignores spatial complexity previously established as an important element in explaining the chronology and directionality of megafauna extinctions, at least in south-eastern Sahul (*Saltré et al., 2019*). Thus, this enticing, but still unsupported hypothesis that warming and drying conditions were related to intrinsic extinction susceptibility, will require more precise estimates of extinction timing and hindcasted climate conditions, and perhaps greater sample sizes across more species.

