## [Decision Letter]

**Acceptance summary:**

This manuscript deploys novel ecological modelling to test hypotheses about the extinction of Sahul's megafauna. It estimates the demographic parameters of several extinct groups, and then introduces perturbations that disrupt these populations in various ways (simulating the effects of changing variables such as ecological conditions, human hunting, and predator-prey relationships). The authors then compare the predictions of when different taxa should go extinct under each scenario to current estimates of the extinction windows of these groups, and find that no single scenario explains the overall chronology of extinction. This is valuable information for palaeontologists, archaeologists, and other palaeo-scientists, but also has implications for understanding biodiversity loss in the present day.

**Decision letter after peer review:**

Thank you for submitting your article "Relative demographic susceptibility does not explain the extinction chronology of Sahul's megafauna" for consideration by *eLife*. Your article has been reviewed by 3 peer reviewers, and the evaluation has been overseen by a Reviewing Editor and George Perry as the Senior Editor. The reviewers have opted to remain anonymous.

Summary:

The reviewers agree that this is an interesting and useful contribution for understanding LQ extinctions, and that it is generally well-presented. It shows that the factors that increase extinction risk are de-coupled from the factors that eventually lead to extinction and thus in its timing. However, the reviewers also note that although the modelling approach is novel, it is reliant on datasets that are biased and at times these biases are not well-accounted for. Because much of the conclusions drawn from the modelling could already be drawn from existing records and using literature that is glossed over here, attention to that literature should be improved and the contributions beyond the megafauna debate should be emphasized. Furthermore, the authors should take care to improve clarity in the framing of the models, the presentation and interpretation of results, figures, and discussion. The reviewers have provided extensive and detailed comments about how to go about doing this, divided below by the Reviewing Editor into major issues additional analyses, conceptual issues, framing.

Essential revisions:

1) Additional analyses (no additional data collection). The reviewers had specific concerns about the effects of sampling on the extinction chronology and the influence of body mass on a number of things (recovery potential, life history/demographic correlates, etc). Specifically, the analytical issues that present the biggest problems revolve around sampling uncertainty and body mass correlation. The former could be addressed by introducing some sensitivity tests. These could be directed towards chronological biases (how does removing one date affect the confidence intervals?), as well as geographical sampling biases (how does removing a region affect the trends?). The latter in particular would be important in the claims of a continental trend. It is also possible that biases are a function of taxon sampling. There are an increasing number of small mammal Pleistocene extinctions being recognized in Australia, and it is unclear if these follow the same trends as the megafauna. If so, that would indeed remove the body size issues.

2) Better framing of the five putative drivers of extinctions (these are not considered or glossed over in the present manuscript)):

i. Appears to assume that only human hunting will differentially affect demographically sensitive species. However, novel or extreme climate change can also affect such species (e.g. Selwood, K.E., McGeoch, M.A. and Mac Nally, R., 2015. The effects of climate change and land‐use change on demographic rates and population viability. Biological Reviews, 90(3), pp.837-853.)

ii. This mechanism is predicated on using a modelling result [ref. 25] as data. It also makes the bold claim that species inhabiting certain habitats are less accessible to human hunters without any consideration of the archaeological or modern record on this point (e.g. Roberts, P., Hunt, C., Arroyo-Kalin, M., Evans, D. and Boivin, N., 2017. The deep human prehistory of global tropical forests and its relevance for modern conservation. Nature Plants, 3(8), pp.1-9; Fa, J.E. and Brown, D., 2009. Impacts of hunting on mammals in African tropical moist forests: a review and synthesis. Mammal Review, 39(4), pp.231-264).

iii. Many of the supporting references here do not seem like logical choices for this argument. E.g. [28] refers to coral-reef fishes. Moreover, this hypothesis conflicts with much modern data showing that extinction risk and body size are correlated under climate and environmental change (e.g. Cardillo, M., Mace, G.M., Jones, K.E., Bielby, J., Bininda-Emonds, O.R., Sechrest, W., Orme, C.D.L. and Purvis, A., 2005. Multiple causes of high extinction risk in large mammal species. Science, 309(5738), pp.1239-1241. Liow, L.H., Fortelius, M., Bingham, E., Lintulaakso, K., Mannila, H., Flynn, L. and Stenseth, N.C., 2008. Higher origination and extinction rates in larger mammals. Proceedings of the National Academy of Sciences, 105(16), pp.6097-6102. Tomiya, S., 2013. Body size and extinction risk in terrestrial mammals above the species level. The American Naturalist, 182(6), pp.E196-E214.)

3) More nuanced interpretation of model output.

The major weakness in this manuscript is in the discussion. The authors should be very clear in their discussion that their model does not indicate that demographic factors had no part in extinct events per se, but rather that they don't explain extinction chronology. Extinction chronologies reflect a number of different factors and processes, but they don't take away from the fact that certain life history traits can make a species more likely to go extinct from those factors.

The authors seem to argue that demographics don't explain the megafaunal extinction in the Sahul, but in fact, their results suggest that they do; the only thing demographics by themselves don't explain is the chronology. Extinction risk as determined by demographic susceptibility is highly related to body mass and generation time (which in turn is also affected by body mass) but differential survival (timing of extinction) is determined by factors such as geographic range size, dispersal ability, access to refugia, and behavioral and morphological adaptations against hunting, and the ability to survive catastrophic events. A reiteration of this point would be beneficial to the clarity of this otherwise well written manuscript.

The authors clearly (and elegantly) show that extinct species, which were all large, and had long generation times, had demographic traits that made them more susceptible to extinction. This is evident in figures 3 and 4. However, in the discussion, in lines 301-303, they state that no demographic trends explain the extinction. This is not supported by the results. While the timing of when species go extinct doesn't correlate with demographic susceptibility, the peculiar nature of the extinction-a large size biased extinction-is explained by demographic factors, and is a phenomenon that has been explored in a global analysis by Lyons et al. 2016 Biol. Lett. Therefore, demographic trends DO explain why certain species go extinct, while others survive. The authors should be careful when they say that "that no obvious demographic trends can explain the great Sahul mass extinction event"; instead, they should re-iterate that no obvious demographic trend explains the extinction chronology.

4) More careful discussion of results relative to literature. The authors further go on to suggest that their results suggest that the extinctions were random, but the size-selectivity clearly shows that the extinctions were in fact not random with respect to body size. Their analyses do show that the rate of extinction doesn't exceed background to the same degree that it's been suggested in prior studies, and this is something that researchers need to explore further. Also, the authors raise an important point in lines 309-311 that human hunting could have interacted with demographic susceptibility, something that Lyons et al. 2016 Biol. Lett. show, and the results of the present study should be discussed in light of the 2016 paper.

They also raise an important point in lines 312-320 that behavioral or morphological adaptations may have allowed some seemingly "high risk" species to persist despite anthropogenic pressure. These model "mis-matches" have been reported by Alroy 2001 Science as well in a multispecies overkill simulation. It would be beneficial to discuss the present results within the context of other examples of model mismatches, such as those from Alroy 2001.

In lines 353-358, the authors once again state that their results show no clear relationship between body-mass and demographic disadvantage, despite clearly showing these relationships in Figures 3 and 4, and even stating as much in the beginning of the discussion. The plots clearly show that large bodied taxa were at a demographic disadvantage. There is a difference between explaining why certain taxa go extinct vs. why they go extinct at a certain point in time, and this should be made clear. The authors are correct in stating that demographic factors don't explain the relative extinction chronology, i.e. when species go extinction relative to each other, but they do explain why large species go extinct, and why these extinctions take place after human arrival. Moreover, generation length, which is also correlated with demographic susceptibility, is highly correlated with body mass (Brook and Bowman 2005 Pop. Ecol), once again showing that body mass-related effects do help explain the extinctions.

The authors rightfully point out earlier in the discussion that spatial variation, local climates, ecological interactions, etc. all influence how and why a particular population disappears. Extinction chronologies reflect a number of different factors and processes, but they don't take away from the fact that certain life history traits can make a species more likely to go extinct from those factors. Large proboscideans like mammoths had a high risk of extinction based on life history traits, but managed to survive on island refugia into the mid-Holocene. Similar other examples exist, and show that extinction chronologies can vary vastly.

Therefore, the lack of correlation can be explained by these factors, and the authors need to expand on these in their discussion, perhaps if possible, by giving specific examples. They should be more careful in their discussion by clearly distinguishing drivers of extinction risk, and how these drivers can be de-coupled from timing, but at the same time providing a good explanation for the biological factors leading to the extinction. Here again the authors should consider the work of Brook and Bowman and Lyons et al.

---

## [Author Response]

Essential revisions:1) Additional analyses (no additional data collection). The reviewers had specific concerns about the effects of sampling on the extinction chronology and the influence of body mass on a number of things (recovery potential, life history/demographic correlates, etc). Specifically, the analytical issues that present the biggest problems revolve around sampling uncertainty and body mass correlation. The former could be addressed by introducing some sensitivity tests. These could be directed towards chronological biases (how does removing one date affect the confidence intervals?)

Hidden chronological biases in time series of dated palaeontological specimens will always exist, which are particularly compounded at the limits of radiocarbon dating. That said, we agree that another treatment of the chronologies is warranted to demonstrate that different sets of assumptions do not change our general conclusions about the lack of a chronological signal.

To this end, we have revisited all the dates we derived from Saltré et al. [1], which were predicted regions of extinction-timing agreement among six frequentist models correcting for the Signor-Lipps effect (apart from the two updated extinction dates for mainland *Thylacinus* and *Sarcophilus* from White et al. [2] that have already been recalculated). For the remaining taxa (*Diprotodon*, *Genyornis*, *Megalibgwilia*, *Metasthenurus*, *Palorchestes*, *Phascolonus*, *Procoptodon*, *Protemnodon*, *Sismosthenurus*, *Sthenurus*, *Thylacoleo*, and *Zygomaturus*), we adapted the Gaussian-resampled, inverse-weighted McInerny (GRIWM) approach we designed and tested previously [3, 4] to derive new estimates and uncertainties of extinction time. Here, we jack-knifed the estimates 10,000 times for each taxon, calculating the median and 95% lower and upper confidence bounds.

This approach does not depend on distributional assumptions given that it is a resampling approach [4], even if it often provides wider uncertainty windows (especially for older dates). The main effect of this approach was to push back the extinction date for *Metasthenurus* (from a median of 50 ka to 89 ka), and push forward some of the dates within the other macropodiformes and vombatiforms. However, the net effect when we related these new estimates to any of the different risk metrics (mass, generation length, vital rate-specific extinction risk, or median susceptibility rank) was to maintain our general conclusion of no relationship for the chronology. We therefore did not repeat the supplementary analysis of the climate conditions based on these new extinction estimates.

We have placed these new results in the supplementary information, essentially making a new version of Figure 1 (in new Appendix S8; new Figure S9), Figure 6 (in new Appendix S8; new Figure S10), and Figure 7 (in new Appendix S8; new Figure S11). We also now refer to these additional results in the Discussion section, and have added a description of the approach in this new Appendix and a brief overview in the Methods. We have also included the GRIWM code in the updated Github respository (github.com/cjabradshaw/MegafaunaSusceptibility), as well as the raw dates for calculating the new estimates for each taxon (.csv files).

Methods:

“We also examined the sensitivity of our overall results to uncertainty in extinction estimates by deriving a jack-knifed version of the Gaussian-Resampled, Inverse-Weighted McInerny (GRIWM) model [3, 4] (Supplementary Information Appendix S8).”

Discussion:

“… went extinct (even after considering alternative approaches to calculate the window of extinction – Supplementary Information Appendix 8; Figure S9–S11).”

As well as geographical sampling biases (how does removing a region affect the trends?). The latter in particular would be important in the claims of a continental trend.

We would dearly love to do this. Even our ‘regional’ (south-eastern Sahul) assessment of spatial patterns of extinction [5] required us to lump taxa into one large group because of a lack of spatial replication of dated fossil specimens. Therefore, it is not yet currently possible to generate regional estimates of extinction time at the taxonomic scale of genus yet for Sahul megafauna. We have already indicated this limitation in paragraph six of the Discussion.

It is also possible that biases are a function of taxon sampling. There are an increasing number of small mammal Pleistocene extinctions being recognized in Australia, and it is unclear if these follow the same trends as the megafauna. If so, that would indeed remove the body size issues.

This is true, but the unfortunate reality is that for the small mammals known to have gone extinct in Pleistocene Sahul, there are few or no accurately dated specimens, no estimates for final extinction dates, and much of the research on these species is yet to be published. Thus, although the reviewer is correct in saying size-biased taxon sampling could affect our results, we are unable to include smaller extinct species in our analyses because we do not know when they went extinct. We have partly mitigated this bias by pairing extinct species with the largest extant species from the same functional/taxonomic group in our analyses. Nonetheless, we now also state in the Discussion that this is an issue and point to addressing it in the future:

“There is also an increasing number of small marsupial and placental mammals known to have gone extinct during the Pleistocene in Sahul [e.g., 6, 7]. The inclusion of these smaller species in future demographic susceptibility/chronology of extinction analyses will likely be insightful, but unfortunately estimates of extinction dates – which require at least 8 – 10 reliability dated specimens [3, 4] – are not yet available for these animals.”

2) Better framing of the five putative drivers of extinctions (these are not considered or glossed over in the present manuscript)):i. Appears to assume that only human hunting will differentially affect demographically sensitive species. However, novel or extreme climate change can also affect such species (e.g. Selwood, K.E., McGeoch, M.A. and Mac Nally, R., 2015. The effects of climate change and land‐use change on demographic rates and population viability. Biological Reviews, 90(3), pp.837-853.)

This is a good suggestion for inclusion. We have now reworded our description of scenario (*i*) and updated Figure 1 to include the ways in which climate change can affect demography as described in Selwood *et al.* [8]. We have also modified the main text to include the reference and some of the most-pertinent conclusions from that paper (paragraph immediately before Figure 1 in the Introduction):

“However, there are many other ways that climate change can alter demography (reviewed in [8]). […] An increasing frequency of climate-induced catastrophes can also drive relatively smaller populations toward extinction faster, meaning large-bodied species with smaller populations are potentially more susceptible.”

ii. This mechanism is predicated on using a modelling result [ref. 25] as data. It also makes the bold claim that species inhabiting certain habitats are less accessible to human hunters without any consideration of the archaeological or modern record on this point (e.g. Roberts, P., Hunt, C., Arroyo-Kalin, M., Evans, D. and Boivin, N., 2017. The deep human prehistory of global tropical forests and its relevance for modern conservation. Nature Plants, 3(8), pp.1-9; Fa, J.E. and Brown, D., 2009. Impacts of hunting on mammals in African tropical moist forests: a review and synthesis. Mammal Review, 39(4), pp.231-264).

Perhaps a debatable concept given previous writings in this area for Sahul [9], but considering we cannot examine this mechanism explicitly with our approach, we have decided to remove reference to habitat access here in the Introduction (including removing the last mechanism in Figure 1). However, we maintained the text regarding the hypothesised relative susceptibility to human hunting in terms of mode of travelling (gait) and burrowing behaviour (Discussion).

iii. Many of the supporting references here do not seem like logical choices for this argument. E.g. [28] refers to coral-reef fishes. Moreover, this hypothesis conflicts with much modern data showing that extinction risk and body size are correlated under climate and environmental change (e.g. Cardillo, M., Mace, G.M., Jones, K.E., Bielby, J., Bininda-Emonds, O.R., Sechrest, W., Orme, C.D.L. and Purvis, A., 2005. Multiple causes of high extinction risk in large mammal species. Science, 309(5738), pp.1239-1241. Liow, L.H., Fortelius, M., Bingham, E., Lintulaakso, K., Mannila, H., Flynn, L. and Stenseth, N.C., 2008. Higher origination and extinction rates in larger mammals. Proceedings of the National Academy of Sciences, 105(16), pp.6097-6102. Tomiya, S., 2013. Body size and extinction risk in terrestrial mammals above the species level. The American Naturalist, 182(6), pp.E196-E214.)

In terms of reference 28 (Monaco et al.), the taxon is somewhat irrelevant. Rather, that paper focuses on the mechanisms of dispersal and establishment to new regions of climate suitability. For this reason, we think it is a useful citation.

For the other suggested papers, we have added them to the relevant text in the Introduction (including Figure 1) and the Discussion.

We acknowledge that scenario (*iv*) is in conflict with a widely observed pattern — climate change selects for a faster pace of life history (including faster growth, earlier maturation and decreased adult survival) and leads to smaller maximum adult body size. Indeed, this pattern corresponds to scenario (*i*) in which species with a slower life history (generally larger animals) succumbed first.

On the other hand, Scenario (*iv*) relates to another widely hypothesised pattern: that species with lower dispersal and higher specialisation are more vulnerable to climate change. In this scenario, larger species (which tend to have higher dispersal and less specialisation) are predicted to have later extinction dates (i.e., opposite pattern to *i*). We have now reworded scenario (*i*) to make the distinction between these scenarios clearer.

3) More nuanced interpretation of model output.The major weakness in this manuscript is in the discussion. The authors should be very clear in their discussion that their model does not indicate that demographic factors had no part in extinct events per se, but rather that they don't explain extinction chronology. Extinction chronologies reflect a number of different factors and processes, but they don't take away from the fact that certain life history traits can make a species more likely to go extinct from those factors.The authors seem to argue that demographics don't explain the megafaunal extinction in the Sahul, but in fact, their results suggest that they do; the only thing demographics by themselves don't explain is the chronology. Extinction risk as determined by demographic susceptibility is highly related to body mass and generation time (which in turn is also affected by body mass) but differential survival (timing of extinction) is determined by factors such as geographic range size, dispersal ability, access to refugia, and behavioral and morphological adaptations against hunting, and the ability to survive catastrophic events. A reiteration of this point would be beneficial to the clarity of this otherwise well written manuscript.

We agree that we are testing whether demographic susceptibility explains the chronology, and we did not intend to suggest that demographic variability played no role in the extinctions. That said, we understand now that perhaps we were not clear enough on the distinction between demographic susceptibility to extinction (something we clearly showed) *versus* testing for a relationship between demographic vulnerability and extinction chronology (a hypothesis we specifically rejected).

The relevant paragraphs of the Discussion appear to be mainly the first and last, which we have now updated to read as follows:

First paragraph:

“The megafauna species of Sahul demonstrate demographic susceptibility to extinction largely following expectations derived from threat risk in modern species – species with slower life histories have higher demographic risk to extinction on average [10-13]. […] As different perturbations compromise different aspects of a species’ life history, its relative susceptibility to extinction compared to other species in its community varies in often unpredictable ways.”

last paragraph:

“That we found no clear patterns among the extinct megafauna of Sahul to explain their relative extinction chronology supports the notion that, at least for mammals, extinction risk can be high across all body masses depending on a species’ particular ecology [15], even if relative extinction risk appears to follow allometric expectations [10-13, 16] as we demonstrated clearly here (Figure 3, 5, 7). […] Nonetheless, more spatially and community-dependent models are still needed to provide a more complete picture of the dynamics of Late Pleistocene megafauna extinctions”

We also modified the 3^rd^ paragraph of the Discussion as follows:

“This lack of relationship to the chronology, combined with the result that many of the extant species had, in fact, some of the highest extinction susceptibilities, suggest that no obvious demographic properties can explain the taxon-specific timing within the Sahul extinction event of the Late Pleistocene. […] However, this conclusion does not accord well with the notion that Late Pleistocene megafauna extinctions were non-random and occurred at a much higher pace than background extinction rates [18, 19].”

The authors clearly (and elegantly) show that extinct species, which were all large, and had long generation times, had demographic traits that made them more susceptible to extinction. This is evident in figures 3 and 4. However, in the discussion, in lines 301-303, they state that no demographic trends explain the extinction. This is not supported by the results. While the timing of when species go extinct doesn't correlate with demographic susceptibility, the peculiar nature of the extinction-a large size biased extinction-is explained by demographic factors, and is a phenomenon that has been explored in a global analysis by Lyons et al. 2016 Biol. Lett. Therefore, demographic trends DO explain why certain species go extinct, while others survive. The authors should be careful when they say that "that no obvious demographic trends can explain the great Sahul mass extinction event"; instead, they should re-iterate that no obvious demographic trend explains the extinction chronology.

As indicated in Response #7, we did not intend to give this impression, but agree that our wording needs to be clearer. We have amended this paragraph to:

“This lack of relationship to the chronology, combined with the result that many of the extant species had, in fact, some of the highest extinction susceptibilities, suggest that no obvious demographic properties can explain the taxon-specific timing within the Sahul extinction event of the Late Pleistocene. […] However, this conclusion does not accord well with the notion that Late Pleistocene megafauna extinctions were non-random and occurred at a much higher pace than background extinction rates [18, 19].”

We have also now cited Lyons et al. [20] in the first paragraph of the Discussion.

4) More careful discussion of results relative to literature. The authors further go on to suggest that their results suggest that the extinctions were random, but the size-selectivity clearly shows that the extinctions were in fact not random with respect to body size. Their analyses do show that the rate of extinction doesn't exceed background to the same degree that it's been suggested in prior studies, and this is something that researchers need to explore further. Also, the authors raise an important point in lines 309-311 that human hunting could have interacted with demographic susceptibility, something that Lyons et al. 2016 Biol. Lett. show, and the results of the present study should be discussed in light of the 2016 paper.

We had meant that the *chronology* of the extinctions might have arisen at random, not that the extinctions themselves were random (we later dismiss this suggestion). But we agree that our wording was unclear. We have reworded this to:

“This opens the possibility that the chronology instead reflects either a random set of circumstances – that species succumbed to circumstantial combinations of stressors depending on the local perturbations experienced by particular populations [5, 17] – or even that the chronology is still insufficiently resolved.”

We are uncertain what the reviewer is referring to with respect to extinction *rate*. While we mentioned that previous work [18, 19] concluded that the Sahul megafauna extinction rate was higher than the background rate, we offer no such data or analysis on rates of extinction (i.e., extinctions per unit time); rather, we demonstrate how quickly populations of certain sizes could have gone extinct at increasing magnitudes of perturbations to demographic rates.

For the Lyons et al. [20] conclusions, we have added a new sentence in the paragraph mentioned after the first sentence:

“Indeed, Lyons et al. [20] concluded that either large-bodied mammals were selectively targeted by humans during the Late Quaternary, that these species were relatively more vulnerable to human hunting than smaller-bodied species, or both.”

They also raise an important point in lines 312-320 that behavioral or morphological adaptations may have allowed some seemingly "high risk" species to persist despite anthropogenic pressure. These model "mis-matches" have been reported by Alroy 2001 Science as well in a multispecies overkill simulation. It would be beneficial to discuss the present results within the context of other examples of model mismatches, such as those from Alroy 2001.

This is a good reference for mismatches, although they only describe hypotheses as opposed to modelled mechanisms. We have therefore added the following sentence to the end of the relevant paragraph:

“Similar hypotheses regarding risk-persistence mismatches in multispecies simulations have been proposed for the Late Pleistocene megafauna extinction event in North America [21].”

In lines 353-358, the authors once again state that their results show no clear relationship between body-mass and demographic disadvantage, despite clearly showing these relationships in Figures 3 and 4, and even stating as much in the beginning of the discussion. The plots clearly show that large bodied taxa were at a demographic disadvantage. There is a difference between explaining why certain taxa go extinct vs. why they go extinct at a certain point in time, and this should be made clear. The authors are correct in stating that demographic factors don't explain the relative extinction chronology, i.e. when species go extinction relative to each other, but they do explain why large species go extinct, and why these extinctions take place after human arrival. Moreover, generation length, which is also correlated with demographic susceptibility, is highly correlated with body mass (Brook and Bowman 2005 Pop. Ecol), once again showing that body mass-related effects do help explain the extinctions.

We have completely redrafted this paragraph as explained in Response #7. We have also added the citation to Brook and Bowman [16] at the end of the first sentence in that paragraph.

The authors rightfully point out earlier in the discussion that spatial variation, local climates, ecological interactions, etc. all influence how and why a particular population disappears. Extinction chronologies reflect a number of different factors and processes, but they don't take away from the fact that certain life history traits can make a species more likely to go extinct from those factors. Large proboscideans like mammoths had a high risk of extinction based on life history traits, but managed to survive on island refugia into the mid-Holocene. Similar other examples exist, and show that extinction chronologies can vary vastly.

Good point. We have added the following sentence encapsulating this idea, and added a new citation:

“For example, large proboscideans like mammoths managed to persist well into the Holocene on island refugia despite having a high intrinsic extinction risk [22].”

Therefore, the lack of correlation can be explained by these factors, and the authors need to expand on these in their discussion, perhaps if possible, by giving specific examples. They should be more careful in their discussion by clearly distinguishing drivers of extinction risk, and how these drivers can be de-coupled from timing, but at the same time providing a good explanation for the biological factors leading to the extinction. Here again the authors should consider the work of Brook and Bowman and Lyons et al.

The changes we have made throughout the Discussion suffice to make this distinction clear now, especially with the addition of the suggested references. Thank you.

References:

1. Saltré F, Rodríguez-Rey M, Brook BW, Johnson CN, Turney CSM, Alroy J, et al. Climate change not to blame for late Quaternary megafauna extinctions in Australia. Nature Communications. 2016. doi: 10.1038/ncomms10511.2. White LC, Saltré F, Bradshaw CJA, Austin JJ. High-quality fossil dates support a synchronous, Late Holocene extinction of devils and thylacines in mainland Australia. Biology Letters. 2018. doi: 10.1098/rsbl.2017.0642.3. Bradshaw CJA, Cooper A, Turney CSM, Brook BW. Robust estimates of extinction time in the geological record. Quaternary Science Reviews. 2012;33:14-9.4. Saltré F, Brook BW, Rodríguez-Rey M, Cooper A, Johnson CN, Turney CSM, et al. Uncertainties in dating constrain model choice for inferring extinction time from fossil records. Quaternary Science Reviews. 2015. doi: 10.1016/j.quascirev.2015.01.022.5. Saltré F, Chadoeuf J, Peters KJ, McDowell MC, Friedrich T, Timmermann A, et al. Climate-human interaction associated with southeast Australian megafauna extinction patterns. Nat Comm. 2019;10(1):5311. doi: 10.1038/s41467-019-13277-0.6. Cramb J, Price GJ, Hocknull SA. Short-tailed mice with a long fossil record: the genus *Leggadina* (Rodentia: Muridae) from the Quaternary of Queensland, Australia. PeerJ. 2018;6:e5639. doi: 10.7717/peerj.5639.7. Cramb J, Hocknull S, Webb GE. High diversity Pleistocene rainforest Dasyurid assemblages with implications for the radiation of the dasyuridae. Austral Ecol. 2009;34(6):663-9. doi: 10.1111/j.1442-9993.2009.01972.x.8. Selwood KE, McGeoch MA, Mac Nally R. The effects of climate change and land-use change on demographic rates and population viability. Biol Rev. 2015;90(3):837-53. doi: 10.1111/brv.12136.9. Johnson CN. Determinants of loss of mammal species during the late Quaternary 'megafauna' extinctions: life history and ecology, but not body size. Proc R Soc Lond B. 2002;269(1506):2221-7.10. Purvis A, Gittleman JL, Cowlishaw G, Mace GM. Predicting extinction risk in declining species. Proc R Soc Lond B. 2000;267(1456):1947-52. doi: 10.1098/rspb.2000.1234.11. Tomiya S. Body size and extinction risk in terrestrial mammals above the species level. Am Nat. 2013;182(6):E196-E214. doi: 10.1086/673489.12. Cardillo M, Mace GM, Jones KE, Bielby J, Bininda-Emonds ORP, Sechrest W, et al. Multiple causes of high extinction risk in large mammal species. Science. 2005;309(5738):1239-41. PubMed PMID: ISI:000231395400045.13. Liow LH, Fortelius M, Bingham E, Lintulaakso K, Mannila H, Flynn L, et al. Higher origination and extinction rates in larger mammals. Proc Natl Acad Sci USA. 2008;105(16):6097. doi: 10.1073/pnas.0709763105.14. Sodhi NS, Brook BW, Bradshaw CJA. Causes and consequences of species extinctions. In: Levin SA, editor. The Princeton Guide to Ecology. Princeton, USA: Princeton University Press; 2009. p. 514-20.15. Davidson AD, Hamilton MJ, Boyer AG, Brown JH, Ceballos G. Multiple ecological pathways to extinction in mammals. Proc Natl Acad Sci USA. 2009;106(26):10702. doi: 10.1073/pnas.0901956106.16. Brook BW, Bowman DMJS. One equation fits overkill: why allometry underpins both prehistoric and modern body size-biased extinctions. Popul Ecol. 2005;47:137-41. doi: 10.1007/s10144-005-0213-4.17. Peters KJ, Bradshaw CJA, Chadœuf J, Ulm S, Bird MI, Friedrich T, et al. Landscape of fear explains trade-off between distance to water and human predation for extinct Australian megafauna. Comm Biol. 2020;in press.18. Koch PL, Barnosky AD. Late Quaternary extinctions: state of the debate. Annu Rev Ecol Evol Syst. 2006;37:215-50. doi: 10.1146/annurev.ecolsys.34.011802.132415.19. Johnson CN. Australia's Mammal Extinctions: A 50 000 Year History. Cambridge, United Kingdom: Cambridge University Press; 2006.20. Lyons SK, Miller JH, Fraser D, Smith FA, Boyer A, Lindsey E, et al. The changing role of mammal life histories in Late Quaternary extinction vulnerability on continents and islands. Biol Lett. 2016;12(6):20160342. doi: 10.1098/rsbl.2016.0342.21. Alroy J. A multispecies overkill simulation of the end-Pleistocene megafaunal mass extinction. Science. 2001;292(5523):1893-6. PubMed PMID: ISI:000169200700043.22. Nogués-Bravo D, Rodríguez J, Hortal J, Batra P, Araújo MB. Climate change, humans, and the extinction of the woolly mammoth. PLoS Biol. 2008;6(4):e79. doi: 10.1371/journal.pbio.0060079.23. Pardi MI, Smith FA. Paleoecology in an era of climate change: how the past can provide insights into the future. In: Louys J, editor. Paleontology in Ecology and Conservation. Berlin, Heidelberg: Springer Berlin Heidelberg; 2012. p. 93-116.24. Willis KJ, Birks HJB. What is natural? The need for a long-term perspective in biodiversity conservation. Science. 2006;314(5803):1261. doi: 10.1126/science.1122667.25. Wingard GL, Bernhardt CE, Wachnicka AH. The role of paleoecology in restoration and resource management—the past as a guide to future decision-making: review and example from the Greater Everglades ecosystem, U.S.A. Front Ecol Evol. 2017;5:11. doi: 10.3389/fevo.2017.00011.26. Bradshaw CJA, Ehrlich PR, Beattie A, Ceballos G, Crist E, Diamond J, et al. Underestimating the challenges of avoiding a ghastly future. Frontiers in Conservation Science. 2021;1:9. doi: 10.3389/fcosc.2020.615419.27. Frankham R, Bradshaw CJA, Brook BW. Genetics in conservation management: revised recommendations for the 50/500 rules, Red List criteria and population viability analyses. Biol Conserv. 2014;170(0):56-63. doi: 10.1016/j.biocon.2013.12.036.28. Saltré F, Rodríguez-Rey M, Brook BW, Johnson CN, Turney CSM, Alroy J, et al. Climate change not to blame for late Quaternary megafauna extinctions in Australia. Nat Comm. 2016;7:10511. doi: 10.1038/ncomms10511. PubMed PMID: WOS:000369019400003.29. Strauss D, Sadler PM. Classical confidence intervals and Bayesian probability estimates for ends of local taxon ranges. Math Geol. 1989;21(4):411-27. doi: 10.1007/BF00897326.30. McCarthy MA. Identifying declining and threatened species with museum data. Biol Conserv. 1998;83(1):9-17. doi: 10.1016/S0006-3207(97)00048-7.31. Marshall CR. Confidence intervals on stratigraphic ranges with nonrandom distributions of fossil horizons. Paleobiology. 1997;23(2):165-73.32. Solow AR, Roberts DL, Robbirt KM. On the Pleistocene extinctions of Alaskan mammoths and horses. Proc Natl Acad Sci USA. 2006;103(19):7351-3. doi: 10.1073/pnas.0509480103.33. McInerny GJ, Roberts DL, Davy AJ, Cribb PJ. Significance of sighting rate in inferring extinction and threat. Conserv Biol. 2006;20(2):562-7. doi: 10.1111/j.1523-1739.2006.00377.x.34. Bradshaw CJA, Cooper A, Turney CSM, Brook BW. Robust estimates of extinction time in the geological record. Quat Sci Rev. 2012;33(0):14-9. doi: 10.1016/j.quascirev.2011.11.021.35. Peters KJ, Saltré F, Friedrich T, Jacobs Z, Wood R, McDowell M, et al. FosSahul 2.0, an updated database for the Late Quaternary fossil records of Sahul. Sci Dat. 2019;6(1):272. doi: 10.1038/s41597-019-0267-3.36. Rodríguez-Rey M, Herrando-Perez S, Gillespie R, Jacobs Z, Saltré F, Brook BW, et al. Criteria for assessing the quality of Middle Pleistocene to Holocene vertebrate fossil ages. Quat Geochronol. 2015;30:69-79. doi: 10.1016/j.quageo.2015.08.002. PubMed PMID: WOS:000366538100009.37. Bird MI, Condie SA, O’Connor S, O’Grady D, Reepmeyer C, Ulm S, et al. Early human settlement of Sahul was not an accident. Sci Rep. 2019;9(1):8220. doi: 10.1038/s41598-019-42946-9.38. Bradshaw CJA, Ulm S, Williams AN, Bird MI, Roberts RG, Jacobs Z, et al. Minimum founding populations for the first peopling of Sahul. Nat Ecol Evol. 2019;3:1057-63. doi: 10.1038/s41559-019-0902-6.39. Bradshaw CJA, Johnson CN, Llewelyn J, Weisbecker V, Strona G, Saltré F. Relative demographic susceptibility does not explain the extinction chronology of Sahul’s megafauna. bioRxiv. 2020:2020.10.16.342303. doi: 10.1101/2020.10.16.342303.40. Bradshaw CJA, Norman K, Ulm S, Williams AN, Clarkson C, Chadoeuf J, et al. Stochastic models support rapid peopling of Late Pleistocene Sahul. Nat Comm. 2021. doi: 10.1038/s41467-021-21551-3.41. Shabani F, Ahmadi M, Peters KJ, Haberle S, Champreux A, Saltré F, et al. Climate-driven shifts in the distribution of koala-browse species from the Last Interglacial to the near future. Ecography. 2019;42(9):1587-99. doi: 10.1111/ecog.04530.